# Delay and Cooperation in Nonstochastic Linear Bandits

**Shinji Ito**
NEC Corporation
i-shinji@nec.com

**Daisuke Hatano**
RIKEN AIP
daisuke.hatano@riken.jp

**Hanna Sumita**
Tokyo Institute of Technology
sumita@c.titech.ac.jp

**Kei Takemura**
NEC Corporation
kei_takemura@nec.com

**Takuro Fukunaga**
Chuo University, RIKEN AIP, JST PRESTO
fukunaga.07s@g.chuo-u.ac.jp

**Naonori Kakimura**
Keio University
kakimura@math.keio.ac.jp

**Ken-ichi Kawarabayashi**
National Institute of Informatics
k-keniti@nii.ac.jp

## Abstract

This paper offers a nearly optimal algorithm for online linear optimization with delayed bandit feedback. Online linear optimization with bandit feedback, or nonstochastic linear bandits, provides a generic framework for sequential decision-making problems with limited information. This framework, however, assumes that feedback can be observed just after choosing the action, and, hence, does not apply directly to many practical applications, in which the feedback can often only be obtained after a while. To cope with such situations, we consider problem settings in which the feedback can be observed $d$ rounds after the choice of an action, and propose an algorithm for which the expected regret is $\tilde{O}(\sqrt{m(m+d)T})$, ignoring logarithmic factors in $m$ and $T$, where $m$ and $T$ denote the dimensionality of the action set and the number of rounds, respectively. This algorithm achieves nearly optimal performance, as we are able to show that arbitrary algorithms suffer the regret of $\Omega(\sqrt{m(m+d)T})$ in the worst case. To develop the algorithm, we introduce a technique we refer to as *distribution truncation*, which plays an essential role in bounding the regret. We also apply our approach to cooperative bandits, as studied by Cesa-Bianchi et al. [18] and Bar-On and Mansour [12], and extend their results to the linear bandits setting.

## 1 Introduction

Bandit linear optimization (nonstochastic linear bandits) models various sequential decision-making problems under partial-information conditions and has a wide range of applications including combinatorial bandits [17] and adaptive routing problems [11]. In this model, a player is given a set $\mathcal{A} \subseteq \mathbb{R}^m$ of actions, each of which corresponds to an $m$-dimensional feature vector, and chooses an action $a_t \in \mathcal{A}$ in each round $t \in [T]$. Just after choosing action $a_t$, the player gets feedback $\ell_t^\top a_t$ of the loss for the chosen action, where $\ell_t \in \mathbb{R}^m$ is a *loss vector*. The goal of the player is to minimize cumulative loss $\sum_{t=1}^T \ell_t^\top a_t$. A number of studies have proposed algorithms for this model, achieving sublinear regret bounds of $\tilde{O}(m\sqrt{T})$ [17; 22; 31].

However, in many applications, we cannot always get feedback *right after* choosing actions, as noted, e.g., in [5]. For example, in the application of advertisement optimization [3; 6; 33], it takes a certain

Table 1: Regret bounds for nonstochastic bandit problems with/without delay

| | Multi-armed bandit ($K$-arms) | Linear bandit ($m$: dim. of action set) |
|---|---|---|
| Without delay | $O(\sqrt{KT\log K})$ [10] | $O(m\sqrt{T\log m})$ [15; 17; 22] |
| | $O(\sqrt{KT})$ [8] | |
| | $\Omega(\sqrt{KT})$ [10] | $\Omega(m\sqrt{T})$ [21] |
| With delay | $O(\sqrt{dKT})$ [35; 36] | $\tilde{O}(m\sqrt{dT})$ [19] |
| ($d$-rounds delay) | $O(\sqrt{(d+K)T\log K})$ [18] | ([19] applies to more general models) |
| | $O(\sqrt{(d\log K + K)T})$ [46] | $\tilde{O}(\sqrt{m(d+m)T})$ **[Theorem 1]** |
| | $\Omega(\sqrt{(d\log K + K)T})$ [18] | $\Omega(\sqrt{m(d+m)T})$ **[Theorem 2]** |

amount of time until the advertisements begin to affect consumers' behavior. Previous studies are not directly applicable to such situations. For the multi-armed bandit problems, algorithms that work well even for delayed-feedback settings have been proposed, as shown in Table 1, but they are not available for more general linear bandits settings. Vernade et al. [43] provided algorithms for *stochastic* linear bandits with delayed feedback, and they work well under assumptions of time-invariant generative models for the loss, without regret bounds for nonstochastic settings.

This paper introduces *bandit linear optimization with delayed feedback*, which includes the multi-armed bandit problems [18], and proposes a min-max optimal algorithm for the model. In this model, there is an additional parameter $d > 1$ representing the rounds of feedback delay. In contrast to the standard bandit feedback model, in which a player observes $\ell_t^\top a_t$ at the end of $t$-th round, in our model, the feedback $\ell_t^\top a_t$ can be observed at the end of the $(t+d)$-th round, i.e., $d$-rounds later.

**Our contribution**

The main contribution of this paper is to construct a nearly-optimal algorithm for online linear optimization with delayed bandit feedback. More precisely, for online linear optimization with an $m$-dimensional action set and $d$-rounds delay, Algorithm 1, described in Section 4, enjoys the following regret bound:

**Theorem 1.** *For arbitrary loss sequences $(\ell_t)_{t=1}^T$, the regret for Algorithm 1 is bounded as*

$$\mathbf{E}[R_T] \le \max\left\{\sqrt{8m(d+em)T\log T} + 3, Cm(d\sqrt{m}+m)\log^3(dmT)\right\},$$

*where $\mathbf{E}[\cdot]$ means the expectation taken w.r.t. the internal randomization of the algorithm and $C > 0$ is a global constant.*

We note that this paper considers *oblivious adversarial model*, i.e., $\ell_t$ is assumed not to depend on the output of the algorithm. Our results can easily be generalized to an adaptive adversarial model, i.e., a model in which $\ell_t$ may depend on $\{a_j\}_{j<t}$. As shown in Table 1, our regret bound improves upon the results presented by Cesa-Bianchi et al. [19]. We should note that their algorithm works for more general settings with *composite* delayed feedback. Their work will be discussed in Section 2.

Our regret bound can be shown to be min-max optimal up to logarithmic factors. In fact, we provide the following regret lower bound:

**Theorem 2.** *Suppose that $\mathcal{A} = \{-1, 1\}^m$ and $\|\ell_t\|_1 \le 1$. There is a distribution of $(\ell_t)_{t=1}^T$ for which an arbitrary algorithm suffers regret as*

$$\mathbf{E}[R_T] = \Omega(\min\{\sqrt{m(d+m)T}, T\}), \tag{1}$$

*where $\mathbf{E}[\cdot]$ means the expectation w.r.t. the randomness of $(\ell_t)$ and the algorithm.*

We show this lower bound by combining the result for bandit linear optimization without delay [21; 26] and for online linear optimization with delayed full-information [44]. This lower bound implies that there is no room to improve the upper bound in Theorem 1, up to logarithmic factors.

Our proposed algorithm is based on the multiplicative weight update (MWU) method [7] with an unbiased estimator $\hat{\ell}_t$ of the loss vector $\ell_t$. MWU methods manage probability distributions $p_t$ over

action set $\mathcal{A}$, and choose an action $a_t$ following $p_t$. Some existing algorithms [17; 22] for bandit linear optimization employ MWU to achieve an optimal regret bound up to logarithmic factors. These algorithms, however, have not been proven to work well for delayed-feedback settings. This issue appears to be due to the behavior of the probability distribution $p_t$. In existing algorithms, $p_t$ is updated using $\hat{\ell}_t$, and $p_t$ may change drastically per round since $\hat{\ell}_t$ is unbounded, which can worsen the regret, especially in delayed-feedback settings.

To deal with the above issue, we employ two techniques to construct a new, more stable unbiased estimator $\hat{\ell}_t$. The first technique is to manage probability distributions over the convex hull $\mathcal{B} :=$ conv$(\mathcal{A})$ of an action set $\mathcal{A}$, rather than $\mathcal{A}$ itself. If we apply MWU to $\mathcal{B}$, the probability distribution $p_t$ will have a property referred to as *log-concavity* [34], which plays an important role in our analysis. The other, and more essential, technique is to *truncate* the distribution, which ensures that the estimator $\hat{\ell}_t$ is bounded in terms of a specific norm depending on the distribution. Additionally, thanks to log-concavity, we can also show that this truncation does not change the distribution drastically. Similar techniques to this truncation can be found in [16; 28], though much difference can be found as well. For example, in contrast to our truncation technique, the focus region introduced in [16] is updated so that the new one is included in the prior one. This property seems essential for stabilizing their kernel-based estimators, but makes the algorithm and the analysis much complicated.

Our algorithm and analysis can be generalized to a multi-agent cooperative bandit setting [12; 18] in which $N$ agents cooperate to solve a common bandit optimization problem while communicating via a network. In this problem, there is an underlying undirected communication graph $G = (V, E)$, each node of which corresponds to an individual agent. Each agent solves a common bandit optimization problem, and the observation of agent $u \in V$ can be shared with another agent $v \in V$ in $d_G(u, v)$-rounds later, where $d_G(u, v)$ denotes the length of the shortest path in $G$ between $u$ and $v$. As shown in Table 2, for cooperative nonstochastic multi-armed bandit problems, Cesa-Bianchi et al. [18] provides an algorithm achieving regret of $\tilde{O}(\sqrt{(1 + \frac{K}{|V|}\alpha(G))T})$ averaged over all agents, where $\alpha(G)$ denotes the independence number of $G$. Bar-On and Mansour [12] achieved $\tilde{O}(\sqrt{(1 + \frac{K}{|N(v)|})T})$-regret for each agent $v$ simultaneously, where $N(v)$ denotes the neighbors of $v$ in $G$. In this paper, we construct an algorithm for a more general linear optimization setting. By combining the techniques in cooperative multi-armed bandit [12; 18] and linear bandits with delayed feedback as in Theorem 1, we obtain an algorithm such that the expected regret of each agent is bounded by $\tilde{O}(\sqrt{m(1 + \frac{m}{|N(v)|})T})$.

**Theorem 3.** *For cooperative nonstochastic linear bandits, there is an algorithm for which the regret of each agent $v$ is bounded as*

$$\mathbf{E}[R_T(v)] \leq \max\left\{16\sqrt{m\left(1 + \log m + \frac{m}{|N(v)|}\right)T\log T} + 3, Cm^2\log^3(mT)\right\}, \quad (2)$$

*where $\mathbf{E}[\cdot]$ means the expectation taken w.r.t. the internal randomization of each agent's algorithm as well as the randomness of loss vectors, and $C$ is a global constant.*

This bound is tight up to logarithmic factors in a special case in which $G$ is a complete graph. Indeed, we provide the following regret lower bound:

**Theorem 4.** *Let $G = (V, E)$ be a complete graph. There is an environment of a cooperative linear bandit problem over $\mathcal{A} = \{-1, 1\}^m$ with the communication graph $G$, for which each agent $v \in V$ suffers regret $R_T(v)$ of at least*

$$\mathbf{E}[R_T(v)] = \Omega\left(\min\left\{\sqrt{m\left(1 + \frac{m}{|V|}\right)T}, T\right\}\right) \quad (3)$$

*for any arbitrary algorithm.*

This lower bound matches the upper bound of Theorem 3 when $|N(v)| = \Omega(|V|)$.

## 2 Related Work

Bandit linear optimization [17; 21; 22] is a generic model for sequential decision-making with partial information. This model includes a well-studied multi-armed bandit problem [10] as a special

Table 2: Regret bounds for cooperative nonstochastic bandit problems

| | Multi-armed bandit ($K$-arms) | Linear bandit ($m$: dim. of action set) |
|---|---|---|
| Upper bound | $\tilde{O}(\sqrt{(1 + \frac{K}{|V|}\alpha(G))T})$ [18] | $\tilde{O}(\sqrt{m(1 + \frac{m}{|N(v)|})T})$ **[Theorem 3]** |
| | $\tilde{O}(\sqrt{(1 + \frac{K}{|N(v)|})T})$ [12] | |
| Lower bound | $\Omega(\sqrt{(\log K + \frac{K}{|V|})T})$ [40] | $\Omega(\sqrt{m(1 + \frac{m}{|V|})T})$ **[Theorem 4]** |

case, where the action set $\mathcal{A} = [K]$ is just a finite set of actions. Other important special cases are *combinatorial bandits* [15; 17], where the action set $\mathcal{A} \subseteq 2^{[K]}$ is a set of subsets of a fixed finite set, and the loss for choosing action $a \in \mathcal{A}$ is given by $\sum_{i \in a} \ell_{ti}$, the sum of losses for the items in $a$. For solving bandit linear optimization, many algorithms have been proposed for stochastic settings [1; 9; 20] as well as for nonstochastic settings [2; 4; 11; 15; 17; 22]. There are known to exist algorithms [15; 19] achieving $\tilde{O}(m\sqrt{T})$-regret, which nearly matches the lower bound of $\Omega(m\sqrt{T})$ shown in [21].

In the context of online optimization, delayed feedback has been considered for a wide range of settings [18; 19; 29; 43; 32; 37; 42; 45; 46] due to its practical significance. As has been noted, e.g., in [18; 29; 44], a regret bound of $R_T \leq U(T)$ for the no-delay setting immediately leads to a bound of $R_T \leq d \cdot U(T/d)$ for the setting with $d$-rounds delay. Our question is, then, how one can achieve a regret bound better than $d \cdot U(T/d)$. For nonstochastic multi-armed bandit settings, some algorithms have been found to achieve better bounds than $d \cdot U(T/d)$ [18; 42; 46], as can be seen in Table 1. For nonstochastic linear bandits, however, such an improvement upon $O(d \cdot U(T/d))$-regret cannot be found in the literature, and this paper offers the first. We should note that Vernade et al. [43] provided an algorithm for linear bandits that works for stochastic settings with delayed feedback. It is also worth noting that Cesa-Bianchi et al. [19] provided a generic framework for reducing bandit problems with delayed and *composite* feedback to those without delay. In their model, the feedback at each step $t$ depends on all chosen actions in the last $d$ steps $t, t-1, \ldots, t-d+1$, i.e., the feedback is expressed as $\sum_{s=0}^{d-1} \ell_{t-s}^{(s)}(a_{t-s})$. This model is applicable to wider problem settings than the delayed feedback model in this paper, as each feedback depends on the chosen action at a single round in the latter model. In their paper, it was shown that any algorithm for a bandit optimization problem without delay can be transformed into one for a counterpart problem with delayed composite feedback.

Multi-agent cooperative bandit online learning has been considered for various settings [11; 12; 14; 19; 25; 30; 38; 40; 41], with applications including, e.g., peer-to-peer recommendation services serving a large number of users connected in a network [13]. While many of these studies are focusing on stochastic models, only a limited number of studies dealing with nonstochastic models can be found. Among them, Cesa-Bianchi et al. [18] have considered a cooperative nonstochastic multi-armed bandit, taking communication delays into account, and proposed an algorithm for which the average (or the sum) of the regret for all agents is bounded. The cooperative bandit model in this paper is based on their model. Bar-On and Mansour [12] provided algorithms for the multi-armed bandit problem in [19], by which the regret for each agent is bounded well. In this paper, we generalize these results to linear bandit problems.

## 3 Problem Settings

### 3.1 Online linear optimization with delayed bandit feedback

A player is given the number $T$ of rounds and an action set $\mathcal{A} \subseteq \mathbb{R}^m$ before the game starts. The action set $\mathcal{A}$ is an arbitrary compact set in $\mathbb{R}^m$, which is not contained in any proper linear subspace. We note that the assumption does not affect the generality of the problem. Indeed, if $\mathcal{A}$ is contained in a proper linear subspace, we can find such a subspace using the linear optimization oracle for $\mathcal{A}$ (e.g., from Corollary 14.1 of [39]). Hence, by reducing the entire vector space into this linear subspace, we can transform the problem so that the assumption holds. In each round $t \in [T]$, the player chooses action $a_t$, and then, if $t > d$, the environment reveals the loss $\ell_{t-d}^\top a_{t-d} \in \mathbb{R}$. Without loss of generality, we assume $d \leq T - 1$. The loss vector $\ell_t \in \mathbb{R}^m$ is assumed to satisfy $|\ell_t^\top a| \leq 1$

for all $a \in \mathcal{A}$. In this paper, we assume that the sequence $(\ell_t)_{t=1}^T$ of loss vectors is an arbitrary *non-adaptive* sequence, i.e., we do not assume any generative model for $\ell_t$, but each $\ell_t$ is assumed not to depend on the output of the algorithm. Player performance is measured by means of *regret $R_T$*, defined as $R_T = \sum_{t=1}^T \ell_t^\top a_t - \min_{a^* \in \mathcal{A}} \sum_{t=1}^T \ell_t^\top a^*$.

### 3.2 Cooperative nonstochastic linear bandits

The model for cooperative bandits in this paper is based on the problem setting considered in [12; 18]. There is a communication graph $G = (V, E)$, an undirected graph each vertex of which corresponds to an agent that plays a common linear bandit problem. In each round $t = 1, 2, \ldots, T$, each agent $v \in V$ chooses an action $a_t(v) \in \mathcal{A}$ from a common action set $\mathcal{A} \subseteq \mathbb{R}^m$, and then observes loss $\ell_t(v)^\top a_t(v) \in [-1, 1]$ for the chosen action, where we assume that $\mathbf{E}[\ell_t(v)] = \ell_t$ for each agent $v$.[1] At the end of each round, each agent $v$ sends a message $m_t(v)$ to neighbors $u \in N(v) = \{u \in V \mid \{u, v\} \in E\}$. The message $m_t(v)$ consists of the chosen action, observed loss, and the distribution $q_t^v$ for choosing an action such that $a_t(v) \sim q_t^v$:

$$m_t(v) = \langle t, v, a_t(v), \ell_t(v)^\top a_t(v), q_t^v \rangle. \tag{4}$$

Note that each agent chooses an action independently, i.e., the action $a_t(v)$ independently follows $q_t(v)$ for each agent $v$. The goal is to construct an algorithm for which the regret $R_T(v) = \sum_{t=1}^T \ell_t^\top a_t(v) - \min_{a^* \in \mathcal{A}} \sum_{t=1}^T \ell_t^\top a^*$ for each agent $v$ is as small as possible.

## 4 Algorithms and Regret Upper Bounds

### 4.1 Preliminary

In this subsection, we introduce a technique that we refer to as *distribution truncation*, which plays a central role in bounding the regret. We denote the convex hull of $\mathcal{A}$ by $\mathcal{B}$. Given a distribution $p$ over $\mathcal{B}$, define $\mu(p) \in \mathbb{R}^m$ and $S(p) \in \mathrm{Sym}(m)$ by $\mu(p) = \mathbf{E}_{x \sim p}[x]$ and $S(p) = \mathbf{E}_{x \sim p}[xx^\top]$. For any vector $x \in \mathbb{R}^m$ and positive semidefinite matrix $A \in \mathrm{Sym}(m)$, denote $\|x\|_A = \|A^{\frac{1}{2}} x\|_2 = \sqrt{x^\top A x}$. Given a distribution $p$ over $\mathcal{B}$, define a *truncated distribution $p'$* by

$$p'(x) = \frac{p(x)\mathbf{1}\{\|x\|_{S(p)^{-1}}^2 \le m\gamma^2\}}{\mathrm{Prob}_{y \sim p}[\|y\|_{S(p)^{-1}}^2 \le m\gamma^2]} \propto p(x)\mathbf{1}\{\|x\|_{S(p)^{-1}}^2 \le m\gamma^2\}, \tag{5}$$

where $\gamma \ge 4\log(10mT)$ is a parameter that we refer to as the *truncation level*. From the definition of $p'$, any sample $b$ chosen from a truncated distribution $p'$ is bounded in terms the norm $\|\cdot\|_{S(p)^{-1}}$, as $\|b\|_{S(p)^{-1}}^2 \le m\gamma^2$. This property ensures that the estimated loss vector (defined in (10)), constructed from a sample from a truncated distribution, has a bounded norm as in (11), thanks to which action distributions do not change drastically per round, as will be shown in Lemmas 5 and 6.

**Properties of log-concave distributions** If a probability distribution has a density function $p : \mathcal{B} \to \mathbb{R}_{\ge 0}$ such that $\log(p(x))$ is a concave function, then we call it a *log-concave* distribution. We use the following concentration property of log-concave distributions:

**Lemma 1.** *If $x$ follows a log-concave distribution $p$ over $\mathbb{R}^m$ satisfying $S(p) \preceq I$, we have*

$$\mathrm{Prob}[\|x\|_2^2 \ge m\alpha^2] \le m\exp(1 - \alpha) \tag{6}$$

*for an arbitrary $\alpha \ge 0$.*

Missing proofs of lemmas are provided in the appendix. From this lemma 1, we can show that $p$ and $p'$ defined as (5) are close if $p$ is a log-concave distribution, in the following sense:

**Lemma 2.** *Suppose that $p$ is a log-concave distribution over $\mathcal{B}$. For any function $f : \mathcal{B} \to [-1, 1]$ and $\gamma \ge 4\log(10mT)$, we have*

$$\left| \mathbf{E}_{x \sim p}[f(x)] - \mathbf{E}_{x \sim p'}[f(x)] \right| \le \frac{1}{T} \quad \text{and} \quad \frac{T}{T+1} \cdot S(p) \preceq S(p') \preceq \frac{T+1}{T} \cdot S(p). \tag{7}$$

**Lemma 3.** *If $y$ follows a one-dimensional log-concave distribution such that $\mathbf{E}[y^2] \le s^2 \le 1/100$, we have*

$$\mathbf{E}[g(y)] \le s^2 + 30 \exp\left(-\frac{1}{s}\right) \le 2s^2 \quad where \quad g(y) = \exp(y) - y - 1. \tag{8}$$

## 4.2 Algorithm for linear bandits with delayed feedback

In our algorithm, we update distribution $p_t$ over $\mathcal{B} := \mathrm{conv}(\mathcal{A})$, by the multiplicative weight update method (MWU) as follows:

$$w_t(x) := \exp\left(-\eta \sum_{j=1}^{t-d-1} \hat{\ell}_j^\top x\right), \quad p_t(x) = \frac{w_t(x)}{\int_{y \in \mathcal{B}} w_t(y)\mathrm{d}y}, \tag{9}$$

where $\eta > 0$ is a parameter referred to as the *learning rate*, and $\hat{\ell}_t$ is as defined below. In each round, we pick $b_t \in \mathcal{B}$ from the *truncated distribution* $p_t'$ of $p_t$. We can get a sample from $p_t'$ by iteratively sampling $b \sim p_t$ until $\|b\|_{S(p_t)^{-1}}^2 \le m\gamma^2$. There is a computationally efficient way for sampling $b \sim p_t$ under mild assumptions since $p_t$ is a log-concave distribution. In fact, we can use the techniques developed in [34] to get samples from $p_t$ with polynomial-time computation, given a membership oracle for $\mathcal{B}$. A membership oracle for $\mathcal{B}$ can be constructed from a linear optimization oracle for $\mathcal{A}$, as stated e.g., in [39]. After getting $b_t$, we choose $a_t \in \mathcal{A}$ so that $\mathbf{E}[a_t|b_t] = b_t$. We can compute such an $a_t$ efficiently, given a linear optimization oracle for $\mathcal{A}$. Indeed, as shown in Corollary 14.1g in [39], given $b \in \mathcal{B} = \mathrm{conv}(\mathcal{A})$ we can compute $\lambda_1, \ldots, \lambda_{m+1} \ge 0$ and $c_1, \ldots, c_{m+1} \in \mathcal{A}$ such that $\sum_{i=1}^{m+1} \lambda_i = 1$ and $\sum_{i=1}^{m+1} \lambda_{ti} c_i = b$ via solving linear optimization over $\mathcal{A}$ polynomial times. By setting $a = c_i$ with probability $\lambda_i$, we obtain $\mathbf{E}[a|b] = b$. The algorithm then plays $a_t$ at the $t$-th round, and the feedback $\ell_t^\top a_t$ will be observed at the end of the $(t + d)$-th round. We define $\hat{\ell}_t$ by

$$\hat{\ell}_t = \ell_t^\top a_t S(p_t')^{-1} b_t. \tag{10}$$

We note that $S(p_t')$ is invertible, which can be concluded from the assumption that $\mathcal{A}$ is not contained in any proper linear subspace. Under this assumption, indeed, $\mathcal{B} = \mathrm{Conv}(\mathcal{A})$ is a full-dimensional convex set with a positive Lebesgue measure. It follows from this fact and Lemma 1 that the domain of $p_t'$ is full-dimensional as well. Thus, the distribution $p_t'$ has a density function taking positive values over a full-dimensional set, which implies that the matrix $S(p_t')$ is invertible. A similar argument can be found, e.g., in [27] (between Eq. (4) and (5)), and is implicitly used in [16] as well. Further, we can compute $S(p_t')$ efficiently. In fact, since $p_t'$ is a log-concave distribution, for any $\varepsilon > 0$, we can calculate an $\varepsilon$-approximation of $S(p_t')$ w.h.p. using $(d/\varepsilon)^{O(1)}$ samples generated from $p_t'$, as can be seen from Corollary 2.7 of [34].

The vector $\hat{\ell}_t$ defined as (10) is an unbiased estimator of $\ell_t$ and bounded in terms of the norm $\|\cdot\|_{S(p_t)}^2$, i.e., we have

$$\mathbf{E}\left[\hat{\ell}_t\right] = \ell_t, \quad \left\|\hat{\ell}_t\right\|_{S(p_t)}^2 \le 4m\gamma^2. \tag{11}$$

In fact, we have $\mathbf{E}[\hat{\ell}_t] = \mathbf{E}\left[S(p_t')^{-1} b_t a_t^\top \ell_t\right] = \mathbf{E}\left[S(p_t')^{-1} b_t b_t^\top \ell_t\right] = \mathbf{E}\left[S(p_t')^{-1} S(p_t') \ell_t\right] = \ell_t$, which means the first part in (11) holds. To show the second part in (11), we use $\|b_t\|_{S(p_t)^{-1}}^2 \le m\gamma^2$, which is ensured by the fact that $b_t$ is sampled from the truncated distribution $p_t'$. It follows that $\|\hat{\ell}_t\|_{S(p_t)}^2 = (\ell_t^\top a_t)^2 \|S(p_t')^{-1} b_t\|_{S(p_t)}^2 \le 2\|S(p_t')^{-1} b_t\|_{S(p_t')}^2 = 2\|b_t\|_{S(p_t')^{-1}}^2 \le 4\|b_t\|_{S(p_t)^{-1}}^2 \le 4m\gamma^2$, where we use the second part of (7) in the first and the second inequalities. The inequality in (11) for $\hat{\ell}_t$ will be used in the analysis of MWU. The procedure can be summarized in Algorithm 1.

Let us next show that Algorithm 1 enjoys the regret bound given in Theorem 1. Since we have $\mathbf{E}[a_t|p_t'] = \mathbf{E}[b_t|p_t'] = \mu(p_t')$, the regret can be bounded as

$$\mathbf{E}[R_T] = \mathbf{E}\left[\sum_{t=1}^{T} \ell_t^\top(a_t - a^*)\right] = \mathbf{E}\left[\sum_{t=1}^{T} \ell_t^\top(\mu(p_t') - a^*)\right]$$

$$= \mathbf{E}\left[\sum_{t=1}^{T} \ell_t^\top(\mu(p_t) - a^*) + \sum_{t=1}^{T} \ell_t^\top(\mu(p_t') - \mu(p_t))\right] \le \mathbf{E}\left[\sum_{t=1}^{T} \ell_t^\top(\mu(p_t) - a^*)\right] + 1, \tag{12}$$

---

**Algorithm 1** An algorithm for online linear optimization with delayed bandit feedback

---

**Require:** Action set $\mathcal{A}$, parameters $T$, $d \leq T - 1$
1: Set $\gamma = 4 \log(10mT)$ and $\eta = \min \left\{ \sqrt{\frac{m \log T}{2(d+em)T}}, \frac{1}{100\gamma^2(d\sqrt{m}+m)} \right\}$.
2: Define $w_1 : \mathcal{B} \to \mathbb{R}_{>0}$ by $w_1(x) = 1$ for all $x \in \mathcal{B}$.
3: **for** $t = 1, 2, \ldots, T$ **do**
4:    Let $p_t$ be a distribution whose density function is proportional to $w_t$ as in (9).
5:    Pick $b_t \sim p'_t$, e.g., by iteratively sampling $b \sim p_t$ until $\|b\|^2_{S(p_t)^{-1}} \leq m\gamma^2$ holds.
6:    If $t > d$, get feedback of $\ell^\top_{t-d} a_{t-d}$, construct an unbiased estimator of $\ell_{t-d}$ as $\hat{\ell}_{t-d} = \ell^\top_{t-d} a_{t-d} \cdot S(p'_{t-d})^{-1} b_{t-d}$, and update $w_t$ by $w_{t+1}(x) = w_t(x) \exp(-\eta \hat{\ell}^\top_{t-d} x)$.
7:    If $t \leq d$, let $w_{t+1} = w_t$.
8: **end for**

---

where $a^* \in \operatorname{argmin}_{a \in \mathcal{A}} \sum_{t=1}^T \ell^\top_t a$ and the last inequality follows from the first part of (7) and the assumption that $\ell^\top_t a \in [-1, 1]$ for all $a \in A$. From this inequality and a standard analysis for continuous multiplicative weight update methods [7; 22; 23], we obtain the following regret bounds:

**Lemma 4.** *If $p_t$ is defined by (9) with $\hat{\ell}_t$ such that $\mathbf{E}[\hat{\ell}_t] = \ell_t$, the regret for $a_t$ is bounded as*

$$\mathbf{E}[R_T] \leq \mathbf{E}\left[ \sum_{t=1}^T \left( \ell^\top_t(\mu(p_t) - \mu(p_{t+d})) + \frac{1}{\eta} \mathop{\mathbf{E}}_{x \sim p_{t+d}} \left[ g(-\eta \hat{\ell}^\top_t x) \right] \right) \right] + \frac{m \log T}{\eta} + 3, \quad (13)$$

*where $g : \mathbb{R} \to \mathbb{R}$ is defined in (8).*

In the following, we give bounds for the right-hand side of (13), separately for the terms $\ell^\top_t(\mu(p_t) - \mu(p_{t+d}))$ and $\mathbf{E}_{x \sim p_{t+d}}\left[ g(-\eta \hat{\ell}^\top_t x) \right]$. The first can be bounded via the following lemma:

**Lemma 5.** *Suppose that $\ell \in \mathbb{R}^m$ satisfies $|\ell^\top a| \leq 1$ for all $a \in \mathcal{A}$ and $\eta \leq 1/(48\gamma^2 m)$. Then, if $p_t$ is defined by (9) with $\hat{\ell}_t$ satisfying (11), it holds for all $t \in [T]$ that $|\mathbf{E}[\ell^\top(\mu(p_t) - \mu(p_{t+1}))]| \leq 2\eta$.*

This lemma can be shown using (11) and Lemma 3. Finally, to bound the term $\mathbf{E}_{x \sim p_{t+d}}\left[ g(-\eta \hat{\ell}^\top_t x) \right]$ in Lemma 4, we use the following lemma:

**Lemma 6.** *Assume $\eta \leq \frac{1}{100\gamma^2(d+1)\sqrt{m}}$. If $p_t$ is defined by (9) with $\hat{\ell}_t$ satisfying (11), for all $t$, we have $S(p_{t+1}) \preceq \left(1 + \frac{1}{d+1}\right) S(p_t)$.*

This lemma follows from (11) and Lemma 1, by induction in $t$. We are now ready to prove Theorem 1.

*Proof of Theorem 1* Combining Lemmas 4 and 5, we have

$$\mathbf{E}[R_T] \leq 2\eta dT + \mathbf{E}\left[ \sum_{t=1}^T \left( \frac{1}{\eta} \mathop{\mathbf{E}}_{x \sim p_{t+d}} \left[ g(-\eta \hat{\ell}^\top_t x) \right] \right) \right] + \frac{m \log T}{\eta} + 3. \quad (14)$$

Let us bound $\mathbf{E}_{x \sim p_{t+d}}\left[ g(-\eta \hat{\ell}^\top_t x) \right]$ using Lemma 3 and (11). We have

$$\mathop{\mathbf{E}}_{x \sim p_{t+d}}\left[ (-\eta \hat{\ell}^\top_t x)^2 \right] = \eta^2 \|\hat{\ell}_t\|^2_{S(p_{t+d})} \leq \eta^2 \left(1 + \frac{1}{d+1}\right)^d \|\hat{\ell}_t\|^2_{S(p_t)} \leq 4e\eta^2 m\gamma^2 \leq \frac{1}{100}, \quad (15)$$

where the first inequality follows from Lemma 6, the second inequality follows from (11), and the last inequality comes from the assumption of $\eta \leq \frac{1}{100\gamma^2(d\sqrt{m}+m)}$. From this inequality and the fact that $\eta \hat{\ell}^\top_t x$ follows a log-concave distribution when $x \sim p_{t+d}$, we have

$$\mathbf{E}\left[ \mathop{\mathbf{E}}_{x \sim p_{t+d}}\left[ g(-\eta \hat{\ell}^\top_t x) \right] \right] \leq 2\,\mathbf{E}\left[ \mathop{\mathbf{E}}_{x \sim p_{t+d}}\left[ (-\eta \hat{\ell}^\top_t x)^2 \right] \right] \leq 2\eta^2 \left(1 + \frac{1}{d+1}\right)^d \mathbf{E}\left[ \left\|\hat{\ell}_t\right\|^2_{S(p_t)} \right],$$

where the first and second inequalities follow from (3) and (15), respectively. From the definition (10) of $\hat{\ell}_t$, we have

$$\mathbf{E}\left[\left\|\hat{\ell}_t\right\|_{S(p_t)}^2\right] = \mathbf{E}\left[(\ell_t^\top a_t)^2 \left\|S(p_t')^{-1}b_t\right\|_{S(p_t)}^2\right] \leq \frac{T+1}{T}\mathbf{E}\left[\left\|S(p_t')^{-1}b_t\right\|_{S(p_t')}^2\right]$$

$$\leq \left(1 + \frac{1}{d+1}\right)S(p_t') \bullet S(p_t')^{-1}S(p_t')S(p_t')^{-1} = \left(1 + \frac{1}{d+1}\right)m,$$

where the first inequality follows from $|\ell_t^\top a_t| \leq 1$ and the second part of (7). Combining the above two inequalities, we obtain $\mathbf{E}\left[\mathbf{E}_{x\sim p_{t+d}}\left[g(-\eta\hat{\ell}_t^\top x)\right]\right] \leq 2\eta^2(1 + \frac{1}{d+1})^{d+1}m \leq 2e\eta^2m$. From this and (14), we have $\mathbf{E}[R_T] \leq 2\eta(d + em)T + \frac{m\log T}{\eta} + 3$. By substituting the parameter setting in Step 1 of Algorithm 1, we obtain the inequality of Theorem 1. $\qquad\square$

## 4.3 Algorithm for cooperative nonstochastic linear bandits

For the cooperative bandit problem, we consider a *center-based algorithm* [12]. A major difference between our algorithm and the one by Bar-On and Mansour [12] is that ours applies to linear bandit settings while theirs focus on multi-armed bandit settings.

Similarly to [12], we first choose *center agents* $C \subseteq V$ and a corresponding *partition* $\{V_c\}_{c\in C}$ of agents with the following properties:

**Theorem 5** (Theorem 12. in [12]). *Given an undirected graph $G = (V, E)$ and a parameter $m \geq 2$, one can find center agents $C \subseteq V$ and a partition $\{V_c\}_{c\in C}$ of $V$ such that the following hold for all $c \in C$: (i) $c \cup N(c) \subseteq V_c$, (ii) subgraphs of $G$ induced by $V_c$ are connected, and (iii) for all $v \in V_c$, it holds that*

$$\frac{\min\{|N(v)|, m\}}{\min\{|N(c)|, m\}} \leq \exp\left(1 - \frac{d_c(v)}{6}\right), \tag{16}$$

*where $d_c(v)$ denotes the distance from $c$ to $v$ in the subgraph of $G$ induced by $V_c$.*

In the center-based algorithm, each center agent $c \in C$ updates its distribution by means of MWU, and the other agents $v \in V_c$ imitate center agent $c$ on the basis of the message $m_t$ in (4). More precisely, for each $c \in C$ we construct breadth-first search tree $T_c$ for the subgraph induced by $V_c$ with root node $c$, and each agent $v \in V_c \setminus \{c\}$ uses the distribution $q_t^v$ defined by $q_t^v = q_{t-1}^{\mathrm{pa}(v)}$, where $\mathrm{pa}(v)$ denotes the parent node of $v$ in $T_c$. Consequently, for each $v \in V_c \setminus \{c\}$, the action distribution $q_t^v$ satisfies $q_t^v = q_{t-d_c(v)}^c$ for $t > d_c(v)$. We define the distribution $q_t^c$ of each center agent $c \in C$ as follows: define a distribution $p_t^c$ over $\mathcal{B} = \mathrm{conv}(\mathcal{A})$ by MWU as follows:

$$w_t^c(x) := \exp\left(-\eta(c)\sum_{j=1}^{t-1}\hat{\ell}_j(c)^\top x\right), \quad p_t^c(x) = \frac{w_t^c(x)}{\int_{y\in\mathcal{B}}w_t^c(y)\mathrm{d}y}, \tag{17}$$

where we set the truncation level $\gamma$ and the learning rate $\eta(c)$ as $\gamma = 4\log(10mT)$ and $\eta(c) = \min\{\frac{1}{4}\sqrt{\frac{m\log T}{T(1 + \log m + m/\min\{|N(c)|, m\})}}, \frac{1}{100\gamma^2m}\}$. Pick $b_t(c)$ from the truncated distribution $p_t^{c\prime}$ defined by (5), and pick $a_t(c)$ so that $\mathbf{E}[a_t(c)|b_t(c)] = b_t(c)$. The distribution $q_t^c$ is defined to be the distribution that $a_t(c)$ follows. The estimated loss $\hat{\ell}_t(c)$ is defined on the basis of the messages $m_t$ (4) from neighbors $N(c)$ of $c$. Each center agent $c \in C$ computes $\hat{\ell}_t(c)$ using $a_t(v)$ and $\ell_t(v)^\top a_t(v)$ for $v \in N(v)$ as follows:

$$\hat{\ell}_t(c) = \frac{1}{|N(c)|}\sum_{v\in N(c)}\ell_t(v)^\top a_t(v)S(p_{t-1}^c{}')^{-1}b_t'(v), \tag{18}$$

where $b_t'(v)$ is chosen from the posterior probability distribution for $b_t(v)$ given $a_t(v)$, i.e., $\mathrm{Prob}[b_t'(v) = b] \propto \mathrm{Prob}[a_t(v)|b_t(v) = b] \cdot p_{t-1}^c{}'(b)$. Combining Lemmas 3, 4, 5, 6 and Theorem 5, we obtain the regret bounds in Theorem 3. The proof of Theorem 3 is given in Section B in the appendix.

# 5 Lower Bounds

In this section, we briefly describe how we can prove regret lower bounds in Theorems 2 and 4. Inspired by the proof of the lower bound for the multi-armed bandit problem with delayed feedback given in [18], we prove lower bounds by combining existing bounds for different settings such as linear bandits without delay [21; 26] and full-information online optimization with delay [44].

To prove Theorem 2, we combine lower bounds of $\Omega(m\sqrt{T})$ and of $\Omega(\sqrt{mdT})$. We note that the first one comes from a lower bound for linear bandits without delay, shown, e.g., in [21]. The second one is also a lower bound for online linear optimization with delayed feedback. As shown in [44], if an online optimization problem without delay admits a regret lower bound of $\Omega(L(T))$, a counterpart with $d$-round delayed feedback has an $\Omega(dL(T/d))$-lower bound. Since there is a lower bound of $\Omega(\sqrt{mT})$ for the online linear optimization (see, e.g., Theorem 3.2. in [24]), we have $\Omega(d\sqrt{mT/d}) = \Omega(\sqrt{mdT})$.

So the only question left is that both the regret bounds of $\Omega(m\sqrt{T})$ and of $\Omega(\sqrt{mdT})$ must come from the same instance. To this end, we construct the problem instance over $\mathcal{A} = \{-1, 1\}^m$ such that $\|\ell_t\|_1 \leq 1$. By considering a mixed distribution of loss vectors, we obtain a lower bound of $\Omega(m\sqrt{T} + \sqrt{mdT}) = \Omega(\sqrt{m(d+m)T})$. A complete proof is provided in Section C in the appendix.

Theorem 4 for cooperative linear bandits can be, again, obtained by combining lower bounds of $\Omega(\sqrt{mT})$ and of $\Omega(m\sqrt{T/|V|})$. The first one comes from a lower bound for full-information online linear optimization, as cooperative bandits are at least harder than full-information online optimization. The second can come from an $\Omega(m\sqrt{T})$-lower bound for signle-player linear bandits [21]. Since we can regard cooperative bandits as a harder version of single-player bandits with $(T \cdot |V|)$ rounds, the sum of regrets over all agents is at least $\Omega(m\sqrt{T \cdot |V|})$, which implies that the regret per agent is of $\Omega(m\sqrt{T/|V|})$. A complete proof of Theorem 4 to show how we construct the problem instance that gives both the regret bounds of $\Omega(\sqrt{mT})$ and of $\Omega(m\sqrt{T/|V|})$ is given in Section D in the appendix.

# 6 Conclusion

In this paper, we considered online linear optimization with $d$-round-delayed bandit feedback, where $d$ was a given parameter fixed for all rounds. We provided an algorithm that achieves nearly-tight regret bounds, and extended this result to the cooperate bandit setting.

An important future work would be to extend the model to deal with the unknown and round-dependent delay as in [46]. We believe that an adaptive way of tuning parameters, such as $\alpha, \eta$ and $\gamma$ in our algorithms, would work well for this general setting. Another future direction is to improve practical computational cost. The proposed algorithms in this paper rely on continuous relaxation and sampling from log-concave distributions, which causes a large computational time in practice, though they run in polynomial time.

## Broader Impact

The authors believe that this paper presents neither ethical nor societal issues, as this is a theoretical work.

## Acknowledgments and Disclosure of Funding

This work was supported by JST, ERATO, Grant Number JPMJER1201, Japan. SI was supported by JST, ACT-I, Grant Number JPMJPR18U5, Japan. TF was supported by JST, PRESTO, Grant Number JPMJPR1759, Japan. NK and KK were supported by JSPS, KAKENHI, Grant Number JP18H05291, Japan.

## Footnotes

[1] In previous studies [18; 12], all players share a common loss vector $\ell_t$, i.e., the loss vectors $\ell_t(v)$ are equal to $\ell_t$ for all $v$. This case is a special case of our problem setting.

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
