[Supplementary Material]

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

# A Proofs of Lemmas

## A.1 Proof of Lemma 1

*Proof.* Since a linear transformation of a log-concave random variable follows a log-concave distribution as well (Theorem 5.1 in [34]), each $x_i$ follows a log-concave distribution and we have $\mathbf{E}[x_i^2] \leq 1$ from the assumption of $S(p) \preceq I$. Hence, we have

$$\text{Prob}[\|x\|_2^2 \geq m\alpha^2] \leq \text{Prob}[\exists i \in [m], x_i^2 \geq \alpha^2] \leq \sum_{i=1}^{m} \text{Prob}[|x_i| \geq \alpha] \leq m\exp(1-\alpha), \quad (19)$$

where the last inequality follows from Lemma 5.7 in [34]. □

## A.2 Proof of Lemma 2

*Proof.* From the definition 5 of $p'$, we have

$$\begin{aligned}
\mathop{\mathbf{E}}_{x\sim p'}[f(x)] &= \frac{1}{\text{Pr}_{x\sim p}[\|x\|_{S(p)^{-1}}^2 \leq m\gamma^2]} \int_{x\in\mathcal{B}} f(x)\mathbf{1}\{\|x\|_{S(p)^{-1}}^2 \leq m\gamma^2\}p(x)\mathrm{d}x \\
&= \frac{1}{1-\delta} \int_{x\in\mathcal{B}} f(x)\mathbf{1}\{\|x\|_{S(p)^{-1}}^2 \leq m\gamma^2\}p(x)\mathrm{d}x \\
&= \frac{1}{1-\delta}\left(\mathop{\mathbf{E}}_{x\sim p}[f(x)] - \int_{x\in\mathcal{B}} f(x)\mathbf{1}\{\|x\|_{S(p)^{-1}}^2 > m\gamma^2\}p(x)\mathrm{d}x\right),
\end{aligned}$$

where we denote $\delta = \text{Pr}_{x\sim p}[\|x\|_{S(p)^{-1}}^2 > m\gamma^2]$. From this expression, we have

$$\begin{aligned}
\left|\mathop{\mathbf{E}}_{x\sim p}[f(x)] - \mathop{\mathbf{E}}_{x\sim p'}[f(x)]\right| &= \frac{1}{1-\delta}\left|-\delta\mathop{\mathbf{E}}_{x\sim p}[f(x)] + \int_{x\in\mathcal{B}} f(x)\mathbf{1}\{\|x\|_{S(p)^{-1}}^2 > m\gamma^2\}p(x)\mathrm{d}x\right| \\
&\leq \frac{1}{1-\delta}\left(\delta\mathop{\mathbf{E}}_{x\sim p}[1] + \int_{x\in\mathcal{B}} \mathbf{1}\{\|x\|_{S(p)^{-1}}^2 > m\gamma^2\}p(x)\mathrm{d}x\right) = \frac{2\delta}{1-\delta},
\end{aligned} \quad (20)$$

where the inequality follows from the assumption that $f(x) \in [-1, 1]$. The value $\delta$ can be bounded via Lemma 1. In fact, when $x$ follows $p$, a log-concave distribution, $S(p)^{-\frac{1}{2}}x$ follows a log-concave distribution as well. In addition, we have $\mathbf{E}[S(p)^{-\frac{1}{2}}xx^\top S(p)^{-\frac{1}{2}}] = S(p)^{-\frac{1}{2}}S(p)S(p)^{-\frac{1}{2}} = I$. Hence, from Lemma 1, we have

$$\delta = \mathop{\text{Pr}}_{x\sim p}[\|x\|_{S(p)^{-1}}^2 > m\gamma^2] = \mathop{\text{Pr}}_{x\sim p}[\|S(p)^{-\frac{1}{2}}x\|_2^2 > m\gamma^2] \leq m\exp(1-\gamma) \leq 3m\exp(-\gamma) \leq \frac{1}{6T}, \quad (21)$$

where the last inequality follows from $\gamma \geq 4\log(10mT)$. Combining (20) and (21), we obtain the first part of (7). We next show the second part of (7). For any $y \in \mathbb{R}^d$, we have

$$\begin{aligned}
y^\top S(p')y &= \mathop{\mathbf{E}}_{x\sim p'}\left[(y^\top x)^2\right] = \frac{1}{1-\delta}\mathop{\mathbf{E}}_{x\sim p}\left[(y^\top x)^2\mathbf{1}\{\|x\|_{S(p)^{-1}}^2 \leq m\gamma^2\}\right] \\
&\leq \frac{1}{1-\delta}\mathop{\mathbf{E}}_{x\sim p}\left[(y^\top x)^2\right] = \frac{1}{1-\delta}y^\top S(p)y.
\end{aligned}$$

Since this holds for all $y \in \mathbb{R}^d$ and $\frac{1}{1-\delta} \leq \frac{T+1}{T}$, the last inequality in (7) holds. Furthermore, we have

$$\begin{aligned}
y^\top S(p)y - y^\top S(p')y &= \mathop{\mathbf{E}}_{x\sim p}\left[(y^\top x)^2\right] - \frac{1}{1-\delta}\mathop{\mathbf{E}}_{x\sim p}\left[(y^\top x)^2\mathbf{1}\{\|x\|_{S(p)^{-1}}^2 \leq m\gamma^2\}\right] \\
&\leq \mathop{\mathbf{E}}_{x\sim p}\left[(y^\top x)^2\mathbf{1}\{\|x\|_{S(p)^{-1}}^2 > m\gamma^2\}\right] \\
&\leq y^\top S(p)y\mathop{\mathbf{E}}_{x\sim p}\left[\|x\|_{S(p)^{-1}}^2\mathbf{1}\{\|x\|_{S(p)^{-1}}^2 > m\gamma^2\}\right], \quad (22)
\end{aligned}$$

where the last inequality follows from the Cauchy–Schwarz inequality:

$$(y^\top x)^2 = \left((S(p)^{\frac{1}{2}}y)^\top S(p)^{-\frac{1}{2}}x\right)^2 \le \|S(p)^{\frac{1}{2}}y\|_2^2 \cdot \|S(p)^{-\frac{1}{2}}x\|_2^2 = y^\top S(p)y \cdot \|x\|_{S(p)^{-1}}^2.$$

The right-hand side of (22) can be bounded by using Lemma 1 as follows:

$$\underset{x\sim p}{\mathbf{E}}\left[\|x\|_{S(p)^{-1}}^2 \mathbf{1}\{\|x\|_{S(p)^{-1}}^2 > m\gamma^2\}\right]$$

$$\le \sum_{n=1}^{\infty}(n+1)^2 m\gamma^2 \underset{x\sim p}{\Pr}\left[n^2 m\gamma^2 \le \|x\|_{S(p)^{-1}}^2 \le (n+1)^2 m\gamma^2\right]$$

$$\le \sum_{n=1}^{\infty}(n+1)^2 m\gamma^2 \cdot m\exp(1-n\gamma)$$

$$\le m^2\gamma^2\exp(2-\gamma)\sum_{n=1}^{\infty}(n+1)^2\exp(-n) \le 40m^2\gamma^2\exp(-\gamma) \le \frac{1}{2T}, \qquad (23)$$

where the second inequality follows from Lemma 1, the third inequality comes from $\gamma \ge 1$ and hence $n\gamma \ge n + \gamma - 1$, the forth inequality follows from the fact that $\sum_{i=1}^{\infty}(i+1)^2\exp(-i) = \frac{1-2e+4e^2}{(e-1)^3} \le 5$, and the last inequality follows from the assumption of $\gamma \ge 4\log(10mT)$. Combining (22) and (23), we have

$$y^\top S(p')y \ge \left(1 - \frac{1}{2T}\right)y^\top S(p)y \ge \frac{T}{T+1}y^\top S(p)y.$$

Since this holds for all $y \in \mathbb{R}^m$, we have the second inequality of (7). $\qquad\square$

### A.3 Proof of Lemma 3

*Proof.* From Lemma 1, and $\mathbf{E}[(y/s)^2] \le 1$, we have

$$\mathrm{Prob}[y \ge n] = \mathrm{Prob}\left[\frac{y}{s} \ge \frac{n}{s}\right] \le \exp\left(1 - \frac{n}{s}\right). \qquad (24)$$

Using this and the fact that $g(y) \le y^2$ for $y \le 1$ and that $g(y) \le \exp(y)$ for $y > 1$, we have

$$\mathbf{E}[g(y)] = \mathbf{E}[g(y)\mathbf{1}\{y \le 1\}] + \mathbf{E}[g(y)\mathbf{1}\{y > 1\}] \le \mathbf{E}[y^2\mathbf{1}\{y \le 1\}] + \mathbf{E}[\exp(y)\mathbf{1}\{y > 1\}]$$

$$\le s^2 + \sum_{n=1}^{\infty}\exp(n+1)\mathrm{Prob}[n < y \le n+1] \le s^2 + \sum_{n=1}^{\infty}\exp(n+1)\exp\left(1 - \frac{n}{s}\right)$$

$$= s^2 + \exp(2)\sum_{n=1}^{\infty}\left(\exp\left(1 - \frac{1}{s}\right)\right)^n = s^2 + \frac{\exp(3 - s^{-1})}{1 - \exp(1 - s^{-1})} \le s^2 + 30\exp(-s^{-1}),$$

where the last inequality follows from the assumption of $0 \le s \le 1/10$ and $\frac{\exp(3)}{1-\exp(-9)} \le 30$. From this and the fact that $30\exp(-x^{-1}) \le x^2$ holds for $0 < x \le 1/10$, we obtain $\mathbf{E}[g(y)] \le 2s^2$ $\qquad\square$

### A.4 Proof of Lemma 4

*Proof.* Since we have $\mathbf{E}[a_t|p_t'] = \mu(p_t')$, the expected regret can be bounded as follows:

$$\mathbf{E}[R_T] = \mathbf{E}\left[\sum_{t=1}^{T}\ell_t^\top(a_t - a^*)\right] = \mathbf{E}\left[\sum_{t=1}^{T}\ell_t^\top(\mu(p_t') - a^*)\right]$$

$$= \mathbf{E}\left[\sum_{t=1}^{T}\ell_t^\top(\mu(p_t) - a^*)\right] + \mathbf{E}\left[\sum_{t=1}^{T}\ell_t^\top(\mu(p_t') - \mu(p_t))\right]$$

$$\le \mathbf{E}\left[\sum_{t=1}^{T}\ell_t^\top(\mu(p_t) - a^*)\right] + 1$$

$$= \mathbf{E}\left[\sum_{t=1}^{T}\ell_t^\top(\mu(p_t) - \mu(p_{t+d}))\right] + \mathbf{E}\left[\sum_{t=1}^{T}\ell_t^\top(\mu(p_{t+d}) - a^*)\right] + 1, \qquad (25)$$

where the inequality follows from the first part of (7). Since $\hat{\ell}_t$ is an unbiased estimator of $\ell_t$, i.e., from (11), the second term in (25) can be expressed as

$$\mathbf{E}\left[\sum_{t=1}^{T}\ell_t^\top\left(\mu(p_{t+d})-a^*\right)\right] = \mathbf{E}\left[\sum_{t=1}^{T}\hat{\ell}_t^\top\left(\mu(p_{t+d})-a^*\right)\right]. \tag{26}$$

The right-hand side of (26) can be bounded via a standard analysis for continuous MAB as follows. We have

$$\int_{x\in\mathcal{B}}\exp\left(-\eta\sum_{j=1}^{t}\hat{\ell}_j^\top x\right)\mathrm{d}x = \int_{x\in\mathcal{B}}\exp\left(-\eta\sum_{j=1}^{t-1}\hat{\ell}_j^\top x\right)\exp\left(-\eta\hat{\ell}_t^\top x\right)\mathrm{d}x$$

$$= \int_{x\in\mathcal{B}}\exp\left(-\eta\sum_{j=1}^{t-1}\hat{\ell}_j^\top x\right)\mathrm{d}x \cdot \mathop{\mathbf{E}}_{x\sim p_{t+d}}\left[\exp(-\eta\hat{\ell}_t^\top x)\right],$$

where the second equality follows from the definition (9) of $p_t$. Since this holds for all $t\in[T]$, we have

$$\log\int_{x\in\mathcal{B}}\exp\left(-\eta\sum_{t=1}^{T}\hat{\ell}_t^\top x\right)\mathrm{d}x - \log\int_{x\in\mathcal{B}}1\mathrm{d}x = \sum_{t=1}^{T}\log\mathop{\mathbf{E}}_{x\sim p_{t+d}}\left[\exp(-\eta\hat{\ell}_t^\top x)\right]$$

$$= \sum_{t=1}^{T}\log\mathop{\mathbf{E}}_{x\sim p_{t+d}}\left[1-\eta\hat{\ell}_t^\top x + g(-\eta\hat{\ell}_t^\top x)\right] \leq \sum_{t=1}^{T}\left(-\eta\hat{\ell}_t^\top\mu(p_{t+d}) + \mathop{\mathbf{E}}_{x\sim p_{t+d}}\left[g(-\eta\hat{\ell}_t^\top x)\right]\right), \tag{27}$$

where the second equality follows from the definition of $g(y)=\exp(y)-y-1$, and the inequality holds since we have $\log(1+z)\leq z$ for $z>-1$. We note that this condition $z>-1$ indeed holds since $z$ here can be expressed as $z = \mathbf{E}[-\eta\hat{\ell}_t^\top x + g(-\eta\hat{\ell}_t^\top x)] = \mathbf{E}[\exp(-\eta\hat{\ell}_t^\top x)]-1 > -1$. The left-hand side of (27) can be bounded via an integration over a subset $\mathcal{B}'=\{x=(1-\frac{1}{T})a^*+y \mid y\in\mathcal{B}\}\subseteq\mathcal{B}$, as follows:

$$\int_{x\in\mathcal{B}}\exp\left(-\eta\sum_{t=1}^{T}\hat{\ell}_t^\top x\right)\mathrm{d}x \geq \int_{x\in\mathcal{B}'}\exp\left(-\eta\sum_{t=1}^{T}\hat{\ell}_t^\top x\right)\mathrm{d}x$$

$$= \frac{1}{T^m}\int_{y\in\mathcal{B}}\exp\left(-\eta\sum_{t=1}^{T}\hat{\ell}_t^\top\left(\left(1-\frac{1}{T}\right)a^*+\frac{1}{T}y\right)\right)\mathrm{d}y$$

$$\geq \frac{1}{T^m}\int_{y\in\mathcal{B}}1\mathrm{d}y\cdot\exp\left(-\eta\sum_{t=1}^{T}\hat{\ell}_t^\top\left(\left(1-\frac{1}{T}\right)a^*+\frac{1}{T}\mu(p_0)\right)\right),$$

where the last inequality follows from the convexity of $\exp(z)$ and Jensen's inequality. Combining this and (27), we obtain

$$\sum_{t=1}^{T}\left(-\eta\hat{\ell}_t^\top\mu(p_{t+d}) + \mathop{\mathbf{E}}_{x\sim p_{t+d}}\left[g(-\eta\hat{\ell}_t^\top x)\right]\right) \geq -m\log T - \eta\sum_{t=1}^{T}\hat{\ell}_t^\top\left(\left(1-\frac{1}{T}\right)a^*+\frac{1}{T}\mu(p_0)\right)$$

$$= -m\log T - \eta\sum_{t=1}^{T}\hat{\ell}_t^\top a^* - \frac{\eta}{T}\sum_{j=1}^{T}\hat{\ell}_t^\top\left(\mu(p_0)-a^*\right),$$

and hence, we have

$$\mathbf{E}\left[\sum_{t=1}^{T}\hat{\ell}_t^{\top}\left(\mu(p_{t+d})-a^*\right)\right]$$

$$\leq \frac{1}{\eta}\mathbf{E}\left[\sum_{t=1}^{T}\mathop{\mathbf{E}}_{x\sim p_{t+d}}\left[g(-\eta\hat{\ell}_t^{\top}x)\right]\right]+\frac{m\log T}{\eta}+\frac{1}{T}\sum_{j=1}^{T}\mathbf{E}\left[\hat{\ell}_t^{\top}\left(\mu(p_0)-a^*\right)\right]$$

$$= \frac{1}{\eta}\mathbf{E}\left[\sum_{t=1}^{T}\mathop{\mathbf{E}}_{x\sim p_{t+d}}\left[g(-\eta\hat{\ell}_t^{\top}x)\right]\right]+\frac{m\log T}{\eta}+\frac{1}{T}\sum_{j=1}^{T}\mathbf{E}\left[\ell_t^{\top}\left(\mu(p_0)-a^*\right)\right]$$

$$\leq \frac{1}{\eta}\mathbf{E}\left[\sum_{t=1}^{T}\mathop{\mathbf{E}}_{x\sim p_{t+d}}\left[g(-\eta\hat{\ell}_t^{\top}x)\right]\right]+\frac{m\log T}{\eta}+2,$$

where the equality follows from (11) and the last inequality follows from the assumption of $|\ell_t^{\top}a|\leq 1$ for all $a\in\mathcal{A}$. Combining this, (25) and (26), we obtain the desired inequality in Lemma 4. $\square$

### A.5 Proof of Lemma 5

*Proof.* Lemma 5 holds for $t\leq d$ since $p_t=p_{t+1}$ follows from the definition (9) for this case. We consider the case of $t>d$ in the following. We start by introducing some notations. Define $\alpha>1$ and $\beta\in\mathbb{R}$ by

$$\alpha=\mathop{\mathbf{E}}_{x\sim p_t}\left[\exp\left(-\eta\hat{\ell}_{t-d}^{\top}(x-\mu(p_t))\right)\right],\quad \beta=\mathop{\mathbf{E}}_{x\sim p_t}\left[\ell^{\top}x\cdot g\left(-\eta\hat{\ell}_{t-d}^{\top}(x-\mu(p_t))\right)\right]. \tag{28}$$

We can confirm that $\alpha\geq 1$ by using Jensen's inequality:

$$\alpha=\mathop{\mathbf{E}}_{x\sim p_t}\left[\exp\left(-\eta\hat{\ell}_{t-d}^{\top}(x-\mu(p_t))\right)\right]\geq\exp\left(\mathop{\mathbf{E}}_{x\sim p_t}\left[-\eta\hat{\ell}_{t-d}^{\top}(x-\mu(p_t))\right]\right)=\exp(0)=1.$$

Since $p_{t+1}(x)\propto p_t(x)\exp(-\eta\hat{\ell}_{t-d}^{\top}x)\propto p_t(x)\exp(-\eta\hat{\ell}_{t-d}^{\top}(x-\mu(p_t)))$ from the definition (9) of $p_t$, we can express $p_{t+1}$ as

$$p_{t+1}(x)=\frac{1}{\alpha}p_t(x)\exp\left(-\eta\hat{\ell}_{t-d}^{\top}(x-\mu(p_t))\right). \tag{29}$$

Hence, we have

$$\ell^{\top}\mu(p_{t+1})=\frac{1}{\alpha}\int\ell^{\top}x\cdot p_t(x)\exp\left(-\eta\hat{\ell}_{t-d}^{\top}(x-\mu(p_t))\right)\mathrm{d}x$$

$$=\frac{1}{\alpha}\int\ell^{\top}x\cdot p_t(x)\left(1-\eta\hat{\ell}_{t-d}^{\top}(x-\mu(p_t))+g\left(-\eta\hat{\ell}_{t-d}^{\top}(x-\mu(p_t))\right)\right)\mathrm{d}x$$

$$=\frac{1}{\alpha}\left(\ell^{\top}\mu(p_t)-\eta\ell^{\top}\mathrm{Cov}(p_t)\hat{\ell}_{t-d}+\beta\right) \tag{30}$$

Hence, using (11), we have

$$\left|\mathbf{E}\left[\ell^{\top}(\mu(p_t)-\mu(p_{t+1}))\right]\right|=\left|\mathbf{E}\left[\frac{\eta}{\alpha}\ell^{\top}\mathrm{Cov}(p_t)\hat{\ell}_{t-d}+\left(1-\frac{1}{\alpha}\right)\ell^{\top}\mu(p_t)-\frac{\beta}{\alpha}\right]\right|$$

$$\leq\left|\mathbf{E}\left[\frac{\eta}{\alpha}\ell^{\top}\mathrm{Cov}(p_t)\ell_{t-d}\right]\right|+\left(1-\frac{1}{\alpha}\right)+\frac{|\beta|}{\alpha}\leq\eta+\alpha-1+|\beta|, \tag{31}$$

where the first inequality follows from (11) and $|\ell^{\top}\mu(p_t)|\leq 1$, and the last inequality follows from $\alpha\geq 1$ and

$$\ell^{\top}\mathrm{Cov}(p_t)\ell_{t-d}\leq\sqrt{\|\ell\|_{\mathrm{Cov}(p_t)}^2\|\ell_{t-d}\|_{\mathrm{Cov}(p_t)}^2}\leq\sqrt{\|\ell\|_{S(p_t)}^2\|\ell_{t-d}\|_{S(p_t)}^2}\leq 1.$$

Let us bound $\alpha$ and $\beta$ using Lemma 3. We have

$$\alpha=\mathop{\mathbf{E}}_{x\sim p_t}\left[g\left(-\eta\hat{\ell}_{t-d}^{\top}(x-\mu(p_t))\right)+\left(-\eta\hat{\ell}_{t-d}^{\top}(x-\mu(p_t))\right)+1\right]$$

$$=\mathop{\mathbf{E}}_{x\sim p_t}\left[g\left(-\eta\hat{\ell}_{t-d}^{\top}(x-\mu(p_t))\right)\right]+1.$$

The right-hand side of this can be bounded by means of Lemma 3. Indeed, for any fixed $\hat{\ell}_{t-d}^\top$ and $\mu(p_t)$, when $x$ follows the log-concave distribution $p_t$ then $\eta\hat{\ell}_{t-d}^\top(x - \mu(p_t))$ follows a one-dimensional log-concave distribution as well since any marginal of a log-concave distribution is log-concave (see, e.g., Theorem 5.1 in [34]). Further, we have

$$\mathop{\mathbf{E}}_{x \sim p_t}\left[\left(-\eta\hat{\ell}_{t-d}^\top(x - \mu(p_t))\right)^2\right] = \eta^2\|\hat{\ell}_{t-d}\|^2_{\mathrm{Cov}(p_t)} \leq \eta^2\|\hat{\ell}_{t-d}\|^2_{S(p_t)}$$

$$\leq \mathrm{e}\eta^2\|\hat{\ell}_{t-d}\|^2_{S(p_{t-d})} \leq 4\mathrm{e}\eta^2 m\gamma^2 \leq \frac{1}{100},$$

where the second inequality follows from Lemma 6 the third inequality follows from (11), and the last inequality comes from the assumption of $\eta \leq \frac{1}{48\gamma^2 m}$.[2] Hence, we can apply Lemma 3 to have

$$\alpha - 1 = \mathop{\mathbf{E}}_{x \sim p_t}\left[g\left(-\eta\hat{\ell}_{t-d}^\top(x - \mu(p_t))\right)\right] \leq 8\mathrm{e}\eta^2 m\gamma^2 \leq \frac{\eta}{2}, \tag{32}$$

where the last inequality follows from the assumption of $\eta \leq \frac{1}{48\gamma^2 m}$. Furthermore, since we have $|\ell^\top x| \leq 1$ for $x \in \mathcal{B}$, $\beta$ defined in (28) can be bounded as

$$|\beta| \leq \mathop{\mathbf{E}}_{x \sim p_t}\left[g\left(-\eta\hat{\ell}_{t-d}^\top(x - \mu(p_t))\right)\right] \leq 8\mathrm{e}\eta^2 m\gamma^2 \leq \frac{\eta}{2}.$$

Combining this, (31) and (32), we obtain the desired inequality in Lemma 5. $\qquad\square$

### A.6 Proof of Lemma 6

*Proof.* For $t \leq d$, the inequality in Lemma 6 holds since $p_{t+1} = p_t$ follows from the definition (9) of $p_t$. We show the inequality in 6 for $t > d$ by induction in $t$. We denote

$$\varepsilon = \frac{1}{2(1+d)}. \tag{33}$$

For arbitrary $y \in \mathbb{R}^m$, we have

$$y^\top S(p_{t+1})y = \frac{\mathbf{E}_{x \sim p_t}\left[(y^\top x)^2 \exp\left(-\eta\hat{\ell}_{t-d}^\top(x - \mu(p_t))\right)\right]}{\mathbf{E}_{x \sim p_t}\left[\exp\left(-\eta\hat{\ell}_{t-d}^\top(x - \mu(p_t))\right)\right]}$$

$$\leq \mathop{\mathbf{E}}_{x \sim p_t}\left[(y^\top x)^2 \exp\left(-\eta\hat{\ell}_{t-d}^\top(x - \mu(p_t))\right)\right]$$

$$= \mathop{\mathbf{E}}_{x \sim p_t}\left[(y^\top x)^2 \exp\left(-\eta\hat{\ell}_{t-d}^\top(x - \mu(p_t))\right)\mathbf{1}\left\{\exp\left(-\eta\hat{\ell}_{t-d}^\top(x - \mu(p_t))\right) \leq 1 + \varepsilon\right\}\right]$$

$$+ \mathop{\mathbf{E}}_{x \sim p_t}\left[(y^\top x)^2 \exp\left(-\eta\hat{\ell}_{t-d}^\top(x - \mu(p_t))\right)\mathbf{1}\left\{\exp\left(-\eta\hat{\ell}_{t-d}^\top(x - \mu(p_t))\right) > 1 + \varepsilon\right\}\right]$$

$$\leq (1 + \varepsilon)y^\top S(p_t)y + \mathop{\mathbf{E}}_{x \sim p_t}\left[(y^\top x)^2 \exp\left(-\eta\hat{\ell}_{t-d}^\top(x - \mu(p_t))\right)\mathbf{1}\left\{-\eta\hat{\ell}_{t-d}^\top(x - \mu(p_t)) > \frac{\varepsilon}{2}\right\}\right], \tag{34}$$

where the first inequality follows from Jensen's inequality, and the last inequality holds as $\exp(y) > 1 + \varepsilon$ implies $y > \varepsilon/2$ for $0 < \varepsilon < 1/2$. Let us evaluate the second term in (34), using Lemma 1. When $x \sim p_t$, we have

$$\mathop{\mathbf{E}}_{x \sim p_t}[(y^\top x)^2] = \|y\|^2_{S(p_t)}.$$

Futhermore, we have

$$\mathop{\mathbf{E}}_{x \sim p_t}[(-\eta\hat{\ell}_{t-d}^\top(x - \mu(p_t)))^2] \leq \eta^2\|\hat{\ell}_{t-d}\|^2_{S(p_t)} \leq \left(1 + \frac{1}{d+1}\right)^d \eta^2\|\hat{\ell}_{t-d}\|^2_{S(p_{t-d})}$$

$$\leq \mathrm{e}\eta^2\|\hat{\ell}_{t-d}\|^2_{S(p_{t-d})} \leq 4\mathrm{e}\eta^2 m\gamma^2$$

where the first inequality follows from $\mathbf{E}[(x - \mu(p_t))(x - \mu(p_t))^\top] \preceq S(p_t)$, the second inequality follows from the inductive assumption and the second part of (7), and the forth inequality follows from the second part of (11). From Lemma 1, we have

$$\operatorname*{Prob}_{x \sim p_t} \left[ \frac{|\eta \hat{\ell}_{t-d}^\top (x - \mu(p_t))|}{2\eta\gamma\sqrt{em}} + \frac{|y^\top x|}{\|y\|_{S(p_t)}} > \alpha \right]$$

$$\leq \operatorname*{Prob}_{x \sim p_t} \left[ \frac{|\eta \hat{\ell}_{t-d}^\top (x - \mu(p_t))|}{2\eta\gamma\sqrt{em}} > \frac{\alpha}{2} \right] + \operatorname*{Prob}_{x \sim p_t} \left[ \frac{|y^\top x|}{\|y\|_{S(p_t)}} > \frac{\alpha}{2} \right] \leq 2 \exp\left(1 - \frac{\alpha}{2}\right) \quad (35)$$

for any $\alpha > 0$. Using this, we have

$$\operatorname*{\mathbf{E}}_{x \sim p_t} \left[ (y^\top x)^2 \exp\left(-\eta \hat{\ell}_{t-d}^\top (x - \mu(p_t))\right) \mathbf{1}\left\{-\eta \hat{\ell}_{t-d}^\top (x - \mu(p_t)) > \frac{\varepsilon}{2}\right\} \right]$$

$$\leq \operatorname*{\mathbf{E}}_{x \sim p_t} \left[ (y^\top x)^2 \exp\left(-\eta \hat{\ell}_{t-d}^\top (x - \mu(p_t))\right) \mathbf{1}\left\{ \frac{|\eta \hat{\ell}_{t-d}^\top (x - \mu(p_t))|}{2\eta\gamma\sqrt{em}} + \frac{|y^\top x|}{\|y\|_{S(p_t)}} > \frac{\varepsilon}{4\eta\gamma\sqrt{em}}\right\} \right]$$

$$\leq \frac{\varepsilon^2 \|y\|_{S(p_t)}^2}{16e\eta^2\gamma^2 m} \sum_{n=1}^\infty (n+1)^2 \exp\left(\frac{(n+1)\varepsilon}{2}\right) \operatorname*{Prob}_{x \sim p_t}\left[ \frac{n\varepsilon}{4\eta\gamma\sqrt{em}} < \frac{|\eta \hat{\ell}_{t-d}^\top (x - \mu(p_t))|}{2\eta\gamma\sqrt{em}} + \frac{|y^\top x|}{\|y\|_{S(p_t)}} \leq \frac{(n+1)\varepsilon}{4\eta\gamma\sqrt{em}}\right]$$

$$\leq \frac{\varepsilon^2 \|y\|_{S(p_t)}^2}{8e\eta^2\gamma^2 m} \sum_{n=1}^\infty (n+1)^2 \exp\left(\frac{(n+1)\varepsilon}{2}\right) \exp\left(1 - \frac{n\varepsilon}{8\eta\gamma\sqrt{em}}\right)$$

$$\leq \frac{\|y\|_{S(p_t)}^2}{2\eta^2\gamma^2 m} \sum_{n=1}^\infty \exp\left((n+1)\varepsilon\right) \exp\left(-\frac{n\varepsilon}{8\eta\gamma\sqrt{em}}\right) = \frac{\exp(\varepsilon)\|y\|_{S(p_t)}^2}{2\eta^2\gamma^2 m} \sum_{n=1}^\infty \exp\left(\varepsilon - \frac{\varepsilon}{8\eta\gamma\sqrt{em}}\right)^n$$

$$\leq \frac{e\|y\|_{S(p_t)}^2}{2\eta^2\gamma^2 m} \sum_{n=1}^\infty \exp\left(-\frac{\varepsilon}{16\eta\gamma\sqrt{em}}\right)^n \leq \frac{e\|y\|_{S(p_t)}^2}{\eta^2\gamma^2 m} \exp\left(-\frac{\varepsilon}{16\eta\gamma\sqrt{em}}\right) \leq \|y\|_{S(p_t)^2} \cdot \varepsilon,$$

where the third inequality follows from (35), the forth inequality follows from $(\frac{\varepsilon(n+1)}{2})^2 \leq \exp(\frac{\varepsilon(n+1)}{2})$, and the forth, fifth and the last inequalities follow from (33) and the assumption of $\eta \leq \frac{1}{100(d+1)\gamma^2\sqrt{m}}$ with $\gamma = 4\log(10mT)$. Combining this and (34), we have

$$y^\top S(p_{t+1})y \leq (1 + 2\varepsilon)y^\top S(p_{t+1})y = \left(1 + \frac{1}{d+1}\right)y^\top S(p_{t+1})y.$$

Since this holds for any $y \in \mathbb{R}^d$, we have the inequality in Lemma 6. $\qquad\square$

## B   Proof of Theorem 3

We start by showing the following properties of $\hat{\ell}_t(c)$ defined in (18):

**Lemma 7.** *The vector $\hat{\ell}_t(c)$ defined by (18) satisfies*

$$\mathbf{E}\left[\hat{\ell}_t(c)\right] = \ell_t, \quad \left\|\hat{\ell}_t(c)\right\|_{S(p_{t-1}^c)}^2 \leq 4m\gamma^2, \quad \mathbf{E}\left[\left\|\hat{\ell}_t(c)\right\|_{S(p_{t-1}^c)}^2\right] \leq \frac{4m}{\min\{|N(c)|, m\}}. \quad (36)$$

*Proof.* Since $b_t'(v)$ is chosen from the posterior of $b_t(v)$ given the observation $a_t(v)$, $b_t'(v)$ and $b_t(v)$ are identically distributed conditioned on $a_t(v)$. Hence, we have

$$\mathbf{E}\left[b_t'(v)a_t(v)^\top | p_{t-1}^c{}'\right] = \mathbf{E}\left[b_t(v)a_t(v)^\top | p_{t-1}^c{}'\right] = \mathbf{E}\left[b_t(v)b_t(v)^\top | p_{t-1}^c{}'\right] = S(p_{t-1}^c{}') \quad (37)$$

for all $v \in N(c)$. From this, we can show the first equality in (36) as follows:

$$\mathbf{E}\left[\hat{\ell}_t(c)\right] = \frac{1}{|N(c)|} \sum_{v \in N(c)} \mathbf{E}\left[S(p_{t-1}^c{}')^{-1} b_t'(v)a_t(v)^\top \ell_t(v)\right]$$

$$= \frac{1}{|N(c)|} \sum_{v \in N(c)} \mathbf{E}\left[\ell_t(v)\right] = \frac{1}{|N(c)|} \sum_{v \in N(c)} \ell_t = \ell_t. \quad (38)$$

The second part in (36) can be shown similarly to the second part of (11): From the definition (18) of $\hat{\ell}_t(c)$, we have

$$\left\|\hat{\ell}_t(c)\right\|_{S(p_{t-1}^c)} \leq \frac{1}{|N(c)|} \sum_{v \in N(c)} \left\|\ell_t(v)^\top a_t(v) S(p_{t-1}^c{}')^{-1} b_t'(v)\right\|_{S(p_{t-1}^c)}$$

$$\leq \frac{1}{|N(c)|} \sum_{v \in N(c)} \left\|S(p_{t-1}^c{}')^{-1} b_t'(v)\right\|_{S(p_{t-1}^c)} \tag{39}$$

where the last inequality follows from the assumption of $|\ell_t(v)^\top a_t(v)| \leq 1$. For all $v \in N(c)$, we have

$$\|S(p_{t-1}^c{}')^{-1} b_t'(v)\|_{S(p_{t-1}^c)}^2 \leq 2\|S(p_{t-1}^c{}')^{-1} b_t'(v)\|_{S(p_{t-1}^c{}')}^2$$

$$= 2\|b_t'\|_{S(p_{t-1}^c{}')^{-1}} \leq 4\|b_t'\|_{S(p_{t-1}^c)^{-1}} \leq 4m\gamma^2, \tag{40}$$

where the first and second inequalities follow from the second part of Lemma 2, and the last inequality follows from the fact that $b_t'$ is chosen from the posterior for $b_t \sim p_{t-1}^c{}'$, the truncated distribution of $p_{t-1}^c$. Combining (39) and (40), we obtain the second part of (36). To show the last part of (36), we evaluate the variance of $\hat{\ell}_t(c)$. As shown in (38), for all $v \in N(c)$ and any fixed $\ell_t(v)$, we have $\mathbf{E}\left[S(p_{t-1}^c{}')^{-1} b_t'(v) a_t(v)^\top \ell_t(v)\right] = \ell_t(v)$ and $S(p_{t-1}^c{}')^{-1} b_t'(v) a_t(v)^\top \ell_t(v)$ are independent for $v \in N(c)$. Hence, we have

$$\mathbf{E}\left[\left\|\hat{\ell}_t(c)\right\|_{S(p_{t-1}^c)}^2\right] = \frac{1}{|N(c)|^2} \mathbf{E}\left[\left\|\sum_{v \in N(c)} S(p_{t-1}^c{}')^{-1} b_t'(v) a_t(v)^\top \ell_t(v)\right\|_{S(p_{t-1}^c)}^2\right]$$

$$= \frac{1}{|N(c)|^2} \mathbf{E}\left[\left\|\sum_{v \in N(c)} \left(S(p_{t-1}^c{}')^{-1} b_t'(v) a_t(v)^\top \ell_t(v) - \ell_t(v)\right)\right\|_{S(p_{t-1}^c)}^2 + \left\|\sum_{v \in N(c)} \ell_t(v)\right\|_{S(p_{t-1}^c)}^2\right]$$

$$= \frac{1}{|N(c)|^2} \left(\sum_{v \in N(c)} \mathbf{E}\left[\left\|S(p_{t-1}^c{}')^{-1} b_t'(v) a_t(v)^\top \ell_t(v) - \ell_t(v)\right\|_{S(p_{t-1}^c)}^2\right]\right)$$

$$+ \frac{1}{|N(c)|^2} \mathbf{E}\left[\left\|\sum_{v \in N(c)} \ell_t(v)\right\|_{S(p_{t-1}^c)}^2\right]. \tag{41}$$

Since we have $\mathbf{E}\left[S(p_{t-1}^c{}')^{-1} b_t'(v) a_t(v)^\top \ell_t(v)\right] = \ell_t(v)$ for any fixed $\ell_t(v)$, we have

$$\mathbf{E}\left[\left\|S(p_{t-1}^c{}')^{-1} b_t'(v) a_t(v)^\top \ell_t(v) - \ell_t(v)\right\|_{S(p_{t-1}^c)}^2\right] \leq \mathbf{E}\left[\left\|S(p_{t-1}^c{}')^{-1} b_t'(v) a_t(v)^\top \ell_t(v)\right\|_{S(p_{t-1}^c)}^2\right]$$

$$\leq 2\mathbf{E}\left[\left\|S(p_{t-1}^c{}')^{-1} b_t'(v)\right\|_{S(p_{t-1}^c{}')}^2\right] = 2\mathbf{E}\left[\|b_t'(v)\|_{S(p_{t-1}^c{}')^{-1}}^2\right] = 2S(p_{t-1}^c{}') \bullet S(p_{t-1}^c{}')^{-1} = 2m, \tag{42}$$

where the last inequality follows from the second part of Lemma 2 and $|a_t(v)^\top \ell_t(v)| \leq 1$ and the second inequality follows from the fact that $b_t'(v) \sim p_{t-1}^c{}'$ after marginalizing $a_t(v)$ out. Further, the second term in the right-hand side of (41) can be bounded as

$$\mathbf{E}\left[\left\|\sum_{v \in N(c)} \ell_t(v)\right\|_{S(p_{t-1}^c)}^2\right] = \mathbf{E}\left[\mathbf{E}_{x \sim S(p_{t-1}^c)}\left[\left(\left(\sum_{v \in N(c)} \ell_t(v)\right)^\top x\right)^2\right]\right] \leq |N(c)|^2, \tag{43}$$

where the last inequality follows from the assumption of $|\ell_t(v)^\top a| \leq 1$ for all $a \in \mathcal{A}$. Combining (41), (42), and (43), we have

$$\mathbf{E}\left[\left\|\hat{\ell}_t(c)\right\|_{S(p_{t-1}^c)}^2\right] \leq \frac{|N(c)| \cdot 2m}{|N(c)|^2} + \frac{|N(c)|^2}{|N(c)|^2} = \frac{\cdot 2m}{|N(c)|} + 1 \leq \frac{4m}{\min\{|N(c)|, m\}}.$$

$$\square$$

From Lemma 7, we can apply Lemmas 5 and 6 to $p_{t-1}^c$. In fact, $p_{t-1}^c$ is identical to the distribution $p_t$ defined by (9) with $\hat{\ell}_t = \hat{\ell}_t(c)$ and $d = 1$, and the first and second parts of Lemma 7 implies that the conditions in (11) hold. Hence, from Lemmas 5 and 6, we have

$$\mathbf{E}\left[|\ell^\top(\mu(p_t^c) - \mu(p_{t+1}^c))|\right] \le 2\eta, \quad S(p_{t+1}^c) \preceq 2S(p_t^c) \tag{44}$$

for any $t \in [T]$, $c \in C$ and $\ell \in \mathbb{R}^m$ such that $|\ell^\top a| \le 1$ for all $a \in \mathcal{A}$.

Let us evaluate the regret of agent $v \in V_c$. We denote $p_t^v = p_{t-d_c(v)}^c$. From $q_t^v = q_{t-d_c(v)}^c$, we have $\mu(q_t^v) = \mu(p_t^{v\prime})$. Further, since $p_t^v = p_{t-d_c(v)}^c$ is identical to the distribution $p_t$ defined by (9) with $\hat{\ell}_t = \hat{\ell}_t(c)$ and $d = d_c(v)$, from Lemma 4, the regret of $v$ is bounded as

$$\mathbf{E}[R_T(v)]$$

$$\le \mathbf{E}\left[\sum_{t=1}^{T}\left(\ell_t^\top\left(\mu(p_t^v) - \mu(p_{t+d_c(v)}^v)\right) + \frac{1}{\eta(c)}\mathop{\mathbf{E}}_{x \sim p_{t+d_c(v)}^v}\left[g\left(-\eta(c)\hat{\ell}_t(c)^\top x\right)\right]\right)\right] + \frac{m\log T}{\eta(c)} + 3$$

$$= \mathbf{E}\left[\sum_{t=1}^{T}\left(\ell_t^\top\left(\mu(p_{t-d_c(v)}^c) - \mu(p_t^c)\right) + \frac{1}{\eta(c)}\mathop{\mathbf{E}}_{x \sim p_t^c}\left[g\left(-\eta(c)\hat{\ell}_t(c)^\top x\right)\right]\right)\right] + \frac{m\log T}{\eta(c)} + 3. \tag{45}$$

From the first part of (44), we have

$$\mathbf{E}\left[\ell_t^\top\left(\mu(p_{t-d_c(v)}^c) - \mu(p_t^c)\right)\right] = \sum_{i=0}^{d_c(v)-1}\mathbf{E}\left[\ell_t^\top\left(\mu(p_{t-d_c(v)+i}^c) - \mu(p_{t-d_c(v)+i+1}^c)\right)\right]$$

$$\le 2\eta(c)d_c(v). \tag{46}$$

We can bound the term $\mathbf{E}\left[\mathop{\mathbf{E}}_{x \sim p_t^c}\left[g\left(-\eta(c)\hat{\ell}_t(c)^\top x\right)\right]\right]$ in (45) using Lemma 3 and (44). In fact, we can confirm that the assumption of Lemma 3 holds, as follows:

$$\mathop{\mathbf{E}}_{x \sim p_t^c}\left[\left(-\eta(c)\hat{\ell}_t(c)^\top x\right)^2\right] \le \eta(c)^2\|\hat{\ell}_t(c)\|_{S(p_t^c)}^2 \le 2\eta(c)^2\|\hat{\ell}_t(c)\|_{S(p_{t-1}^c)}^2 \le 8\eta(c)^2 m\gamma^2 \le 1/100,$$

where the second and third inequalities follows from (44). Hence, by applying Lemma 3 to $y = -\eta(c)\hat{\ell}_t(c)^\top x$, we have

$$\mathbf{E}\left[\mathop{\mathbf{E}}_{x \sim p_t^c}\left[g\left(-\eta(c)\hat{\ell}_t(c)^\top x\right)\right]\right] \le 2\eta(c)^2\mathbf{E}\left[\|\hat{\ell}_t(c)\|_{S(p_t^c)}^2\right]$$

$$\le 4\eta(c)^2\mathbf{E}\left[\|\hat{\ell}_t(c)\|_{S(p_{t-1}^c)}^2\right] \le \frac{16\eta(c)^2 m}{\min\{|N(c)|, m\}}, \tag{47}$$

where the second inequality follows from the second part of (44), and the last inequality follows from the last part of Lemma 7. Combining (45), (46), and (47), we obtain

$$\mathbf{E}[R_T(v)] \le \eta(c)\left(2d_c(v) + \frac{16m}{\min\{|N(c)|, m\}}\right)T + \frac{m\log T}{\eta(c)} + 3. \tag{48}$$

From the inequality in Theorem 5, we have

$$d_c(v) \le 6\left(1 + \log\frac{\min\{|N(v)|, m\}}{\min\{|N(c)|, m\}}\right) \le 6(1 + \log m), \quad \frac{1}{\min\{|N(c)|, m\}} \le \frac{e}{\min\{|N(v)|, m\}}. \tag{49}$$

Combining (48) and (49), we have

$$\mathbf{E}[R_T(v)] \leq \eta(c) \left( 12(1 + \log m) + \frac{16m}{\min\{|N(c)|, m\}} \right) T + \frac{m \log T}{\eta(c)} + 3$$

$$\leq 16\eta(c) \left( 1 + \log m + \frac{m}{\min\{|N(c)|, m\}} \right) T + \frac{m \log T}{\eta(c)} + 3$$

$$\leq \max \left\{ 8\sqrt{m \left( 1 + \log m + \frac{m}{\min\{|N(c)|, m\}} \right) T \log T}, 100m^2\gamma^2 \log T \right\} + 3$$

$$\leq \max \left\{ 8\sqrt{m \left( 1 + \log m + \frac{em}{\min\{|N(v)|, m\}} \right) T \log T}, 100m^2\gamma^2 \log T \right\} + 3,$$

$$\leq \max \left\{ 16\sqrt{m \left( 1 + \log m + \frac{m}{|N(v)|} \right) T \log T}, 100m^2\gamma^2 \log T \right\} + 3,$$

where the first inequality follows from (48) and the first part of (49) the third inequality follows from the parameter setting of $\eta(c) = \min\{\frac{1}{4}\sqrt{\frac{m \log T}{T(1 + \log m + m/\min\{|N(c)|, m\})}}, \frac{1}{100\gamma^2 m}\}$, and the forth inequality follows from the second part of (49).

## C  Proof of Theorem 2

*Proof of Theorem 2* We first construct a problem instance for which $R_T = \Omega(\sqrt{mdT})$. Let $T$ be a multiple of $md$, i.e., we denote $T = Smd$ with an integer $S$. For each $s = 0, 1, \ldots, S - 1$ and $i = 1, 2, \ldots, m$, we define $\ell_t$ for $t \in [smd + d(i-1) + 1, smd + id]$ by $\ell_t = b_{si}\chi_i$, where $b_{si}$ follows a Bernoulli distribution over $\{-1, 1\}$ with parameter $1/2$ for $s$ and $i$, independently. Since $\ell_t$ and $a_t$ are independent for all $t$ and $\mathbf{E}[\ell_t] = 0$, we have

$$\mathbf{E}\left[ \sum_{t=1}^{T} \ell_t^\top a_t \right] = 0. \tag{50}$$

On the other hand, we also have

$$\mathbf{E}\left[ \min_{a \in \mathcal{A}} \sum_{t=1}^{T} \ell_t^\top a \right] = -\sum_{i=1}^{m} \mathbf{E}\left[ \left| \sum_{t=1}^{T} \ell_{ti} \right| \right] = -d \sum_{i=1}^{m} \mathbf{E}\left[ \left| \sum_{s=1}^{S} b_{si} \right| \right]. \tag{51}$$

Since $\sum_{s=1}^{S} b_{si}$ follows a binomial distribution $Bi(S, 1/2)$, we have $\mathrm{Prob}[|\sum_{s=1}^{S} b_{si}| \geq \sqrt{S}/10] \geq 1/5$. Hence we have

$$\mathbf{E}\left[ \min_{a \in \mathcal{A}} \sum_{t=1}^{T} \ell_t^\top a \right] \leq -\frac{1}{50} dm\sqrt{S}. \tag{52}$$

This implies that $\mathbf{E}[R_T] \geq dm\sqrt{S}/50 = \sqrt{dmT}/50$. Even when $T \leq dm$, we can show $\mathbf{E}[R_T] \geq T$ similarly. Hence, we have

$$\mathbf{E}[R_T] \geq \min\left\{ \frac{\sqrt{dmT}}{50}, T \right\}. \tag{53}$$

We next provide a distribution of $\ell_t$ for which the regret is $\Omega(m\sqrt{T})$. Let $\varepsilon = \min\{\frac{1}{6}, \frac{m}{\sqrt{8T}}\}$. Consider generating $\ell_t$ in the following process: First, pick $a^* \in \{-1, 1\}^d$ uniformly at random. Then for $t = 1, \ldots, T$, pick $i_t \in [m]$ uniformly at random and set $\ell_t = s_t\chi_{i_t}$ where $s_t = a_{i_t}^*$ with probability $\frac{1-\varepsilon}{2}$ and $s_t = -a_{i_t}^*$ with probability $\frac{1+\varepsilon}{2}$. Then, the regret is bounded as

$$\mathbf{E}[R_T] \geq \mathbf{E}\left[ \sum_{t=1}^{T} \ell_t^\top a_t - \sum_{t=1}^{T} \ell_t^\top a^* \right] = \sum_{t=1}^{T} \mathbf{E}\left[ \ell_t^\top a_t - \ell_t^\top a^* \right]. \tag{54}$$

As shown in Lemmas 3 and 4 in [26], we have $\mathbf{E}\left[\ell_t^\top a_t - \ell_t^\top a^*\right] \geq \varepsilon/2$ for any algorithm if $\varepsilon \leq \min\{\frac{1}{6}, \frac{m}{\sqrt{8T}}\}$. Hence, the regret is bounded as

$$\mathbf{E}[R_T] \geq \frac{T\varepsilon}{2} = \min\left\{\frac{m\sqrt{T}}{32}, \frac{T}{12}\right\}. \tag{55}$$

If $(\ell_t)_{t=1}^T$ follows the first distribution with probability $1/2$ and the second distribution with probability $1/2$, the regret is bounded as

$$\mathbf{E}[R_T] \geq \min\left\{\frac{m\sqrt{T}}{64} + \frac{\sqrt{dmT}}{100}, \frac{T}{12}\right\} \geq \min\left\{\frac{\sqrt{m(d+m)T}}{100}, \frac{T}{12}\right\}. \tag{56}$$

$\square$

## D  Proof of Theorem 4

*Proof of Theorem 4* We first construct a problem instance for which $R_T(v) = \Omega(\sqrt{mT})$. Let $T$ be a multiple of $m$, i.e., we denote $T = Sm$ with an integer $S$. For each $s \in [S]$ and $i \in [m]$, we define $\ell_t$ by $\ell_t = b_{si}\chi_i$ for $t = (s-1)m + i$, where $b_{si}$ follows a Bernoulli distribution over $\{-1, 1\}$ with parameter $1/2$ independently for $s \in [S]$ and $i \in [m]$. We set $\ell_t(v) = \ell_t$ for all $v \in V$. Since $\ell_t$ and $a_t(v)$ are independent for all $t$ and $\mathbf{E}[\ell_t] = 0$, we have

$$\mathbf{E}\left[\sum_{t=1}^T \ell_t^\top a_t(v)\right] = 0. \tag{57}$$

On the other hand, we also have

$$\mathbf{E}\left[\min_{a\in\mathcal{A}} \sum_{t=1}^T \ell_t^\top a\right] = -\sum_{i=1}^m \mathbf{E}\left[\left|\sum_{t=1}^T \ell_{ti}\right|\right] = -\sum_{i=1}^m \mathbf{E}\left[\left|\sum_{s=1}^S b_{si}\right|\right]. \tag{58}$$

Since $\sum_{s=1}^S b_{si}$ follows a binomial distribution $Bi(S, 1/2)$, we have $\mathrm{Prob}[|\sum_{s=1}^S b_{si}| \geq \sqrt{S}/10] \geq 1/5$. Hence, we have

$$\mathbf{E}\left[\min_{a\in\mathcal{A}} \sum_{t=1}^T \ell_t^\top a\right] \leq -\frac{1}{50}m\sqrt{S}. \tag{59}$$

This implies that $\mathbf{E}[R_T(v)] \geq m\sqrt{S}/50 = \sqrt{mT}/50$. Even when $T \leq m$, we can show $\mathbf{E}[R_T] \geq T$ similarly. Hence, we have

$$\mathbf{E}[R_T] \geq \min\left\{\frac{\sqrt{dmT}}{50}, T\right\}. \tag{60}$$

We next provide a lower bound of $\Omega(m\sqrt{T/|V|})$. Let $\varepsilon = \min\{\frac{1}{6}, \frac{m}{\sqrt{8T|V|}}\}$. Consider generating $\ell_t(v)$ in the following process: First, pick $a^* \in \{-1, 1\}^d$ uniformly at random. Then for $t = 1, \ldots, T$ and $v \in V$, pick $i_t(v) \in [m]$ uniformly at random and set $\ell_t(v) = s_t(v)\chi_{i_t(v)}$ where $s_t(v) = a^*_{i_t(v)}$ with probability $\frac{1-\varepsilon}{2}$ and $s_t(v) = -a^*_{i_t(v)}$ with probability $\frac{1+\varepsilon}{2}$. Then, as can be shown from Lemmas 3 and 4 in [26], the regret is bounded as

$$\mathbf{E}[R_T(v)] \geq \frac{T\varepsilon}{2} = \min\left\{\frac{m}{32}\sqrt{\frac{T}{|V|}}, \frac{T}{12}\right\}. \tag{61}$$

If $(\ell_t)_{t=1}^T$ follows the first distribution with probability $1/2$ and the second distribution with probability $1/2$, the regret is bounded as

$$\mathbf{E}[R_T] \geq \min\left\{\frac{m}{64}\sqrt{\frac{T}{|V|}} + \frac{\sqrt{mT}}{100}, \frac{T}{12}\right\} \geq \min\left\{\frac{1}{100}\sqrt{m\left(1 + \frac{m}{|V|}\right)T}, \frac{T}{12}\right\}. \tag{62}$$

$\square$