[Reviews · NeurIPS 2020]

Review 1

Summary and Contributions: The paper mainly considers the adversarial linear bandits with delay feedback. The algorithm adopt the distribution truncation technique, which gives the optimal regret bound up to logarithmic factors. Also the algorithm considers the cooperative bandits. Lower bounds corresponding to two settings are also provided.

Strengths: The algorithm is simple and strong, which is promising to apply to different setting.

Weaknesses: 1. It looks like prior work combining cooperative and delay model together, while this paper only considers them separately. 2. The computation issues. Although log-concave distribution can be sample in polynomial time, it still requires more computation time than other discrete linear bandit algorithms, not to mention the algorithm has to sample multiple times per round in order to truncate the original probability distribution. Also, it is not clear to me how to compute S(p')^-1 efficiently.

Correctness: 1. It is not clear to \hat{\ell} is unbiased. The point is that S(p_t')^-1 may not exist. I think a fix is that we have some assumption on the feasible region, but it definitely affects the generality of the algorithm. 2. Moreover, the computation of S(p_t')^-1 can also be problematic. I am not sure it can be calculated efficiently. It is not even a log-concave distribution. 3. In Lemma 4 and equation (27), it is not trivial the inequality can be applied. There should be some condition on eta and gamma. But I guess it is okay as the algorithm truncate the distribution, but I think the authors may need to explain more.

Clarity: Yes

Relation to Prior Work: Yes

Reproducibility: Yes

Additional Feedback: After reading the authors' responses, I think the authors adequately answer my questions. Now I have no concerns about the correctness and the significance of the contribution. I am voting for accepting this paper. However, I hope the authors can add more details on approximation of S(p'_t)^-1 and the time complexity in the final version.


Review 2

Summary and Contributions: The paper considers linear bandits in dimension m with delays d. Trivially one can obtain a bound sqrt(m*m*d*T), where one m comes from measuring the volume of the action set, and the term m*d comes from the variance calculation. The present papers proves that in fact the variance can be m+d, matching a COLT 2016 paper by Cesa-Bianchi et al. for the basic multi-armed bandit case (that is, they had proved K+d for the variance).

Strengths: I find the truncation trick to be very interesting. To my knowledge this is the first time this is applied to the linear bandit setting, and it seems much more natural than the spanners type of approaches. It would be nice to discuss whether it relates to the focus region of [Bubeck, Eldan, Lee, STOC 2017].

Weaknesses: The contribution can be viewed as narrow, but it is not a significant weakness since the problem is also fundamental.

Correctness: I have read some of the proofs and they are correct.

Clarity: The paper is well written.

Relation to Prior Work: In addition to the reference BEL17 that should be discussed, I think in table 1 the no-delay linear bandit bound should be attributed to [15].

Reproducibility: Yes

Additional Feedback:


Review 3

Summary and Contributions: This paper studies the non-stochastic linear bandit problem with a fixed delay horizon d. They provide up to log factors matching upper and lower bounds. They also extend the results on cooperative bandits to the linear setting. Rebuttal update: No change in evaluation.

Strengths: This is the first result for non-stochastic linear bandits with delays. It is highly non-trivial and requires a novel technique to adapt the mwu algorithm.

Weaknesses: The paper only considers the ``easy'' setting of fixed and known delay d for all time steps. In the multi-armed bandit literature, the more advanced problem of an unknown cumulative delay budget D has been studied. It would be nice if the authors mention this line of work and explain the difficulties arising in the linear setting.

Correctness: I did not check the proofs of the Lemmas, but the main body seems fine.

Clarity: Yes.

Relation to Prior Work: Yes.

Reproducibility: Yes

Additional Feedback:

[Author Response · NeurIPS 2020]

**Dear Reviewer #2:**

> prior work combining cooperative and delay model together, while this paper only considers them separately.

Our work combines cooperative and delay models together as well. In the cooperative setting defined in Section 3.2., the message is sent at the end of each round, which implies that the feedback information observed by an agent is transmitted to another distant agent with a delay depending on their distance, as mentioned in Lines 245–248. Therefore, the cooperative setting in our paper includes delayed-feedback problems. This structure of delay and cooperation is the same as the prior work in COLT 2016 paper by Cesa-Bianchi et al.

> 2. it still requires more computation time than other discrete linear bandit algorithms, not to mention ..

For some discrete linear bandit problems, our algorithm is more efficient than existing algorithms. This is because existing algorithms (such as in [Cesa-Bianchi and Lugosi (2012)]) rely on sampling over a combinatorial action set, which can be computationally hard depending on the action set. For example, when the action set is the set of all maximum matchings of a given non-bipartite graph, there is no known polynomial-time algorithm for sampling over this discrete action set. In such a case, our approach, continuous relaxation and truncation, is computationally better. As the reviewer mentions, however, improving practical computational cost is important future work. We shall mention the computational weakness that the reviewer pointed out in the revised version.

> 1. It is not clear to $\hat{\ell}$ is unbiased. The point is that $S(p'_t)^{-1}$ may not exist. I think a fix is that we have ..

We can see that $S(p'_t)$ is invertible from the assumption that $\mathcal{A}$ is not contained in any proper linear subspace, which is stated at Line 140. Under this assumption, indeed, $\mathcal{B} = \mathrm{Conv}(\mathcal{A})$ is a full-dimensional convex set with a positive Lebesgue measure. Combining this and Lemma 1, we can see that the domain of $p'_t$ is full-dimensional as well. Therefore, the distribution $p'_t$ has a density function taking positive values over a full-dimensional convex set, which implies that $S(p'_t)$ is invertible. A similar argument can be found, e.g., in p.8 of [Ito et al., oracle-efficient algorithms for online linear optimization with bandit feedback, NeurIPS2019] (between Eq. (4) and (5)), and is implicitly used in [Bubeck, Eldan, Lee, STOC2017] as well. In the revised manuscript, we add a more clarified proof for this fact. We also would like to note that the assumption at Line 140 does not affect the generality of the problem. Indeed, if $\mathcal{A}$ is contained in a proper linear subspace, we can find such a subspace using the linear optimization oracle for $\mathcal{A}$ (e.g., from Corollary 14.1 of [Schrijver (1998)]). Hence, by reducing the entire vector space into this linear subspace, we can transform the problem so that the assumption holds.

> 2. Moreover, the computation of $S(p'_t)^{-1}$ can also be problematic. .. It is not even a log-concave distribution.

We can see that $p'_t$ is a log-concave distribution as its domain $\{x \in \mathbb{R}^m \mid \|x\|^2_{S(p)^{-1}} \leq m\gamma^2\} \cap \mathcal{B}$ is a convex region and the density function $p_t$ defined by (9) is log-concave. Since $\tilde{p}_t$ is log-concave, for any $\epsilon > 0$, we can get an $\epsilon$-approximation of $S(\tilde{p}_t)$ w.h.p. by generating $(d/\epsilon)^{O(1)}$ samples from $\tilde{p}_t$, from Corollary 2.7 of [Lovàsz and Vempara (2007)]. Samples from $\tilde{p}_t$ can be generated with their polynomial-time sampling algorithm as mentioned around Line 188. A similar discussion can be found in Lemma 5.17 of [Bubeck, Lee, Eldan (2017)]. This fact is used in [Hazan and Karnin (2016)] as well. We shall clarify this in the revised manuscript.

> 3. In Lemma 4 and equation (27), it is not trivial the inequality can be applied.

As the reviewer pointed out, in (27), we need to confirm that $y > -1$ holds for applying the inequality $\log(1 + y) \leq y$. This condition $y > -1$ indeed holds since $y$ can be expressed as $y = \mathbf{E}[-\eta\hat{\ell}_t^\top x + g(-\eta\hat{\ell}_t^\top x)] = \mathbf{E}[\exp(-\eta\hat{\ell}_t^\top x)] - 1 > -1$. We add a more clarified explanation in the revised manuscript.

**Dear Reviewer #3:**

> It would be nice to discuss whether it relates to the focus region of [Bubeck, Eldan, Lee, STOC 2017].

Thanks for providing an important reference. Their technique is similar to ours in that they truncate the domain using the covariance matrix, though much difference can be found as well. For example, in contrast to our truncation technique, their focus region is updated so that the new one is included in the prior one. This property seems essential for stabilizing their kernel-based estimators, but makes the algorithm and the analysis much complicated. In the revised paper, we cite this reference and add a discussion on the relation to this.

**Dear Reviewer #4:**

> The paper only considers the "easy" setting of fixed and known delay d for all time steps.

As the reviewer pointed out, it is a significant future work to extend the model to deal with the unknown and round-dependent delay. We shall mention this in the revised manuscript. We believe that adjusting parameters adaptively should work well for this general setting. However, we have not yet found such a sophisticated way of parameter setting.

[Meta-Review · NeurIPS 2020]

This paper presents the first optimal (up to logarithmic factors) regret bound for the adversarial linear bandit problem with delayed feedback (with a fixed delay). Using this approach, results on cooperative bandits are also extended to the linear bandit setting. All reviewers were very positive about the paper. (Beside taking into account the suggestions of the reviewers, in the final version, please move the Broder Impact section to its designated place.)